# Optimized Graph Structures for Calibrating Graph Neural Networks with Out-of-Distribution Nodes

## Abstract

Graph neural networks (GNNs) have achieved remarkable success in tasks such as node classification, link prediction, and graph classification. However, despite their effectiveness, the reliability of GNN predictions remains a major concern, particularly when graphs contain out-of-distribution (OOD) nodes. To date, the calibration of GNNs in the presence of OOD nodes remains largely underexplored. Our empirical studies reveal that the calibration problem becomes significantly more complex in the presence of OOD nodes, and existing calibration methods are notably less effective in such scenarios. Recently, graph structure learning (GSL), a family of data-centric learning approaches, has shown promise in mitigating the adverse effects of noisy information in graph topology by jointly optimizing the graph structure and GNN training. However, current GSL methods do not explicitly address the calibration challenges posed by OOD nodes. To tackle this challenge, we propose a novel framework called Graph Calibration via Structure Optimization (GCSO) to calibrate GNNs in the presence of OOD nodes. Our empirical findings suggest that reducing the weights of edges connecting in-distribution (ID) and OOD nodes can effectively alleviate the calibration issue. However, identifying such edges and determining their appropriate weights is challenging due to the unknown distribution of OOD nodes. To address this, GCSO introduces an iterative edge-sampling mechanism that captures the topological information of the graph and formulates the structure learning process as a Markov Decision Process (MDP). We then leverage an actor–critic method to dynamically adjust edge weights and evaluate their impact on target node predictions. Additionally, we design a tailored reward signal to guide the policy function toward an optimal adaptive graph structure that minimizes the influence of OOD nodes. Notably, the optimized graph structure can be seamlessly integrated with existing temperature scaling–based calibration techniques for further performance gains. Experimental results on benchmark datasets demonstrate that our method significantly reduces the expected calibration error while maintaining competitive accuracy.

## 1 Introduction

Graph neural networks have proven to be effective for processing graph-structured data, which is prevalent in real-world applications such as social networks, traffic systems, and financial networks. Despite their remarkable success in tasks like node classification, link prediction, and graph classification, the reliability of GNNs has become an increasing concern within the machine learning community. A foundational study (Guo et al., 2017) introduces the expected calibration error to quantify the discrepancy between a model's predictive confidence and the actual likelihood of correctness. More recent works (Wang et al., 2021; Hsu et al., 2022; Teixeira et al., 2019; Fang et al., 2024; Yang et al., 2024b) have observed that GNNs often suffer from underconfidence in their predictions.

Existing GNN calibration methods (Wang et al., 2021; Hsu et al., 2022; Fang et al., 2024; Tang et al., 2024)primarily focus on *clean* graphs, where all nodes are assumed to come from the same distribution. However, real-world graphs often contain out-of-distribution nodes (Zhao et al., 2020; Yang et al., 2022; Song & Wang, 2022). For instance, users in social networks may be connected not only to family and friends but also to strangers or even online scammers.

When a graph includes OOD nodes, the calibration of GNNs tends to deteriorate. As illustrated in Fig. 1, we designate nodes from specific classes (e.g., the last two classes in Cora (Yang et al., 2016)) as OOD nodes, while treating the remaining nodes as in-distribution nodes. We then perform node classification using GCN (Kipf & Welling, 2016) across several graph benchmark datasets and compare the ECE in scenarios with and without OOD nodes. The results indicate that the presence of OOD nodes leads to an increase in ECE. Moreover, the calibration issue becomes more nuanced in the OOD setting. As shown in Fig. 2, unlike the general underconfidence observed in GNNs on clean graphs (Wang et al., 2021; Hsu et al., 2022), GNNs in OOD scenarios tend to be overconfident on some nodes and underconfident on others. Our experiments

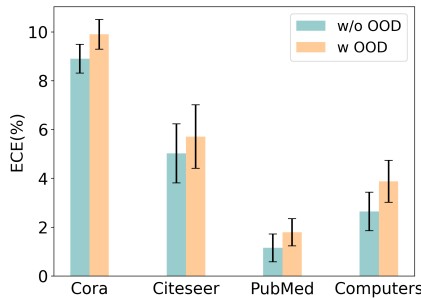

Figure 1: The expected calibration error of GCN on graphs with and without OOD nodes.

further demonstrate that existing graph calibration methods become less effective in the presence of out-of-distribution nodes.

Recently, graph structure learning (Wu et al., 2022; Zou et al., 2023) has shown promising results in mitigating the adverse effects of potential flaws in graph topology, such as redundant, incorrect, or missing connections, by optimizing the graph structure. Yang et al. (Yang et al., 2024a) propose the Data-centric Graph Calibration (DCGC) framework, which reduces calibration error by modifying the graph structure and assigning higher weights to decisive and homophilic edges. However, this approach does not explicitly consider out-of-distribution scenarios. To address calibration in the presence of OOD nodes, Shi et al. (Shi et al., 2023) employ deep Q-learning (Mnih et al., 2013) to calibrate graphs containing OOD nodes. While their method leverages reinforcement learning to modify the graph, it assigns fixed weights to the selected edges, and the values of these weights must be determined through prior knowledge or multiple trials, without accounting for variations in the topological structure. Ideally, edge weights should dynamically adapt based on the distribution of OOD nodes. This limitation may lead to suboptimal calibration performance.

Inspired by prior works (Yang et al., 2024a; Shi et al., 2023) that improve GNN calibration by reweighting edges, we conduct an empirical study and find that calibration issues can be alleviated to some extent by adjusting the weights of edges connected to OOD nodes. However, developing adaptive edge weights that account for the adverse effects of OOD nodes is a non-trivial problem. To address this challenge, we propose a novel framework called Graph Calibration via Structure Optimization (GCSO), which calibrates GNNs through optimized adaptive graph structures. Our approach is built upon the actor–critic paradigm. Specifically, we designate labeled nodes as target nodes and sample their adjacent edges. To guide the selection of informative edges, we introduce a novel iterative strategy based on a predefined discrepancy score, formulating the process as a Markov Decision Process. We then leverage the actor–critic paradigm to dynamically assess the impact of adjusted edge weights on the target nodes. In particular, we adopt Deep Deterministic Policy Gradient (DDPG) (Lillicrap et al., 2016) to generate fine-grained, topology-aware edge weights. In addition, we design a novel reward signal to guide the optimization of edge weights. The reward signal consists of two components: an indicator function and an entropy regularization term for the target nodes. The indicator function aims to preserve node classification accuracy under the optimized graph structure, while the entropy term regularizes the logit distribution of the target nodes for calibration purposes. Notably, the optimized adaptive graph structure can be seamlessly integrated with existing post-hoc calibration methods to further improve calibration performance in downstream tasks. Furthermore, our empirical study suggests that the learned graph structure yields a more distinct edge weight distribution between ID and OOD nodes compared to prior data-centric methods (e.g., DCGC (Yang et al., 2024a)). In addition, the learned edge weights are automatically adaptive to the distribution of OOD nodes, eliminating the need for predefined values as required in prior work (Shi et al., 2023). The contributions of this paper are summarized as follows:

- We propose Graph Calibration via Structure Optimization (GCSO), a graph structure optimization framework for improving calibration under OOD-node settings. Specifically, we formulate graph structure optimization as a Markov Decision Process (MDP) and design a calibration-aware reward

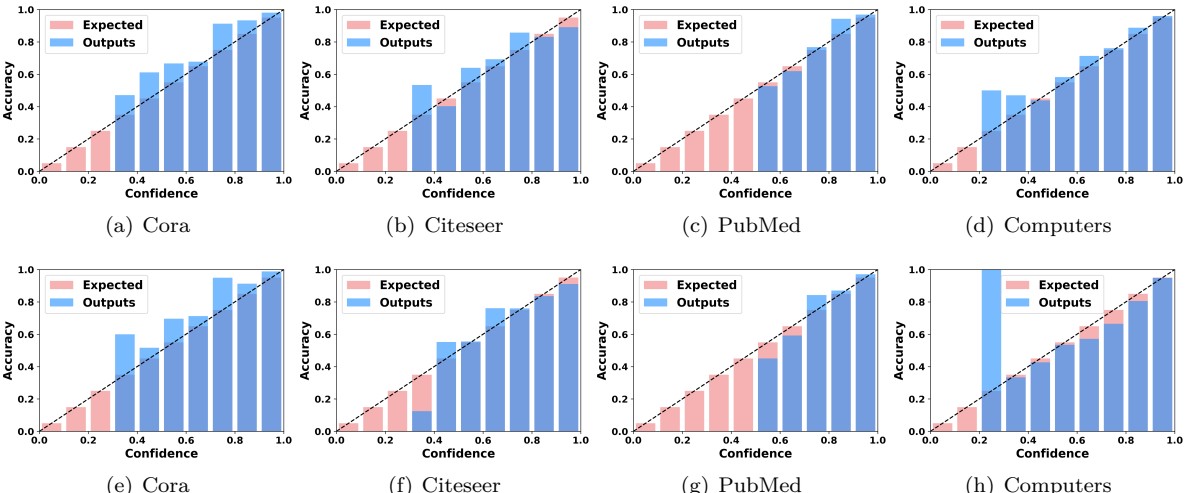

Figure 2: Reliability diagrams of GCN on without OOD nodes ((a)-(d)) and with OOD nodes ((e)-(h)). The results suggest that the calibration issue (i.e., overconfidence or underconfidence) is more complicated with the presence of OOD nodes. Ideally, in a well-calibrated network, accuracy should align with confidence, meaning the height of the blue bars (accuracy) should be as close as possible to the height of the red bars (confidence). When the blue bar is taller than the red bar, it indicates underconfidence. Conversely, if the blue bar is shorter than the red bar, it indicates overconfidence.

  signal to guide the policy function toward learning adaptive graph structures that mitigate the influence of OOD nodes.

- We demonstrate that GCSO learns calibration-friendly graph structures that effectively mitigate the influence of OOD nodes. When combined with existing post-hoc calibration methods, GCSO consistently improves calibration performance and achieves superior results compared with existing graph calibration and graph structure optimization approaches on multiple benchmark datasets.

- Experimental results further reveal that the learned edge weights are transferable, offering benefits in graph learning across various GNN architectures. Specifically, our optimized graph structure can enhance the performance in tasks such as node classification and OOD detection.

## 2 Related Works

**Neural Network Calibration.** The pursuit of developing reliable and trustworthy models has attracted significant attention from researchers, extending into the domain of graph neural networks (GNNs). Guo et al. (Guo et al., 2017) first introduced calibration error to measure the alignment between predictive confidence and accuracy in deep neural networks. Since then, extensive research (Mukhoti et al., 2020; Ghosh et al., 2022; Tao et al., 2023; Wang et al., 2022; 2024; Tang et al., 2024) has been devoted to improving the calibration of neural networks. Recent work by Wang et al. (Wang et al., 2021) applies post-hoc calibration to the logits of GCN (Kipf & Welling, 2016) to obtain calibrated predictions. Uncertainty estimation methods (Lakshminarayanan et al., 2017; Malinin & Gales, 2018) also contribute to improved calibration by modeling the predictive distribution over labels. Wang et al. (Wang et al., 2022) proposed the GCL loss to mitigate the underconfidence issue of GNNs in an end-to-end manner. In addition, GATS (Hsu et al., 2022) was designed to account for influential factors affecting GNN calibration. Fang et al. (Fang et al., 2024) highlighted that the ability of GNNs to distinguish between correct and incorrect predictions is crucial for achieving well-calibrated outcomes, and proposed a simple yet effective discriminative calibration model for GNNs. Tang et al. (Tang et al., 2024) provided theoretical insights into the role of node-wise similarity in GNN calibration and proposed a novel framework that leverages similarity at both global and local levels. Yang et al. (Yang et al., 2024b) identified a key limitation in existing GNN calibration methods, which predominantly focus on the highest logit while ignoring the full spectrum of prediction probabilities. To address

this issue, they proposed the Balanced Calibrated Graph Neural Network (BCGNN), which aims to achieve balanced calibration between overconfidence and underconfidence, supported by solid theoretical justification. Recent graph calibration methods include GETS (Zhuang et al., 2025), which improves calibration through ensemble temperature scaling, and Simple yet efficient graph CAlibRation method (SCAR) (Huang et al., 2025), which calibrates GNNs by adjusting the final prediction layer in a unified and efficient manner. Unlike these methods, our approach focuses on graph structure optimization and is specifically designed to mitigate the impact of OOD-node contamination on calibration. Unlike post-hoc methods that adjust temperature parameters, Yang et al. (Yang et al., 2024a) approached calibration from a data-centric perspective by modifying graph structures and assigning larger weights to decisive and homophilic edges to reduce calibration error. Shi et al. (Shi et al., 2023) further investigated calibration in graphs containing OOD nodes and proposed a framework called GERDQ. However, there are fundamental differences between our work and GERDQ (Shi et al., 2023). First, we introduce a novel edge iteration approach to better capture graph topology. Second, while GERDQ also calibrates graphs by reweighting edges, it assigns fixed weights to adjusted edges without considering structural variations (e.g., the distribution of OOD nodes). In contrast, our method generates fine-grained, adaptive edge weights that adapt to the graph structure. Finally, we design a new reward function to guide the optimization of graph structure, ensuring both improved node classification accuracy and enhanced calibration performance. While these methods improve calibration from different perspectives, they do not explicitly address the simultaneous overconfidence and underconfidence induced by OOD-node contamination. In contrast, our work is motivated by this observation and mitigates its effect through graph structure optimization.

**Graph Structure Learning.** Graph Structure Learning (GSL) aims to address graphs with unreliable, low-quality, or noisy structures—such as redundant or incomplete edges—by learning an optimized topology. To date, extensive research has been conducted in this field. Wu et al. (Wu et al., 2022) introduced a kernelized Gumbel-Softmax operator to efficiently approximate discrete latent structures among data points and proposed a transformer-based model to learn optimal graph topology from node features and labels. To address the lack of robustness and interpretability in existing GSL methods, Zou et al. (Zou et al., 2023) proposed the SE-GSL framework, which explicitly captures the hierarchical semantics of graphs and enhances the robustness of mainstream GNN approaches against noisy and heterophilous structures. To reduce the reliance of GSL methods on label information, Liu et al. (Liu et al., 2022) introduced the Structure Bootstrapping Contrastive Learning framework (SUBLIME), a novel unsupervised approach that leverages self-supervised contrastive learning to optimize graph structure. While prior works mainly focus on clean graphs, our approach aims to learn an optimized, adaptive graph structure for calibrating GNNs in the presence of OOD nodes.

**Reinforcement Learning on Graphs.** The rapid development of reinforcement learning (RL) across various domains has motivated researchers to explore novel RL-based methods for addressing graph-related problems, such as neighborhood detection, information aggregation, and adversarial attacks. GraphNAS (Gao et al., 2019) improves graph learning through neural architecture search by designing a search space of sampling functions, aggregation functions, and gating mechanisms, and using reinforcement learning to identify effective GNN architectures. Unlike GraphNAS, which optimizes the GNN architecture, our method optimizes the graph topology itself through adaptive edge reweighting to learn calibration-friendly graph structures under OOD-node settings. Policy-GNN (Lai et al., 2020) adaptively determines the number of aggregation steps for each node via deep Q-learning (Mnih et al., 2013). RL-Explainer (Shan et al., 2021) and GFlowExplainer (Li et al., 2023) adopt off-policy reinforcement learning methods for graph explanation.

**Graph Learning with OOD.** Most graph learning methods are built on the assumption that training and testing data are independent and identically distributed (i.i.d.). Song et al. (Song & Wang, 2022) first introduced graph learning with OOD nodes and developed the OODGAT framework to jointly perform node classification and OOD node detection. The core idea of OODGAT is to identify OOD nodes and reduce the connections between ID and OOD nodes. Another line of work focuses on graph OOD detection. GNNSAFE (Wu et al., 2023) performs OOD node detection via a learning-free energy-based belief propagation scheme. Yang et al. (Yang et al., 2024c) further proposed NODESAFE to address the susceptibility of GNNSAFE to unbounded negative energy scores and logit shifts. Wang et al. (Wang et al., 2025) utilized a unique implicit adversarial training paradigm for graph OOD detection. In GPN (Stadler et al., 2021),

OOD node detection is achieved through uncertainty estimation. GraphDE (Li et al., 2022), a probabilistic generative framework, jointly performs graph debiased learning and OOD node detection.

## 3 Preliminary

### 3.1 Problem Formulation

We first present the problem formulation of our study. Consider an attributed graph $\mathcal{G} = \{\mathcal{V}, \mathcal{E}, \mathbf{X}\}$, where the finite node set is denoted by $\mathcal{V} = \{i \mid 1 \leq i \leq N\}$, and the edge set is denoted by $\mathcal{E} \subseteq \mathcal{V} \times \mathcal{V}$. Here, $N$ is the total number of nodes in the graph, and the feature matrix is denoted by $\mathbf{X} \in \mathbb{R}^{N \times d}$, where $d$ is the dimensionality of node features. The graph structure can be represented by a binary adjacency matrix $\mathbf{A} \in \{0, 1\}^{N \times N}$. In graph learning with out-of-distribution nodes, the node set can be partitioned into an in-distribution node set and an OOD node set, i.e., $\mathcal{V} = \mathcal{V}_{\text{ID}} \cup \mathcal{V}_{\text{OOD}}$. The features of OOD nodes are sampled from a different distribution than those of ID nodes, i.e., $P(X_{\text{OOD}}) \neq P(X_{\text{ID}})$. The label space for ID nodes is $Y = \{1, 2, \cdots, K\}$, while OOD nodes are assumed not to belong to any of these categories, and their labels are unknown. In semi-supervised graph learning, the ID node set can be further divided into labeled and unlabeled subsets, i.e., $\mathcal{V}_{\text{ID}} = \mathcal{V}_{\text{ID}}^l \cup \mathcal{V}_{\text{ID}}^{ul}$. The goal of standard semi-supervised graph learning is to learn a classifier $f : (\mathbf{X}, \mathbf{A}) \to \tilde{Y}$ that maps node features and graph structure to predicted labels $\tilde{Y}$. However, the presence of unknown OOD nodes makes this task more challenging. Effectively mitigating the negative impact of OOD nodes is therefore a key challenge in semi-supervised graph learning.

In our study, the expected calibration error is used as a primary evaluation metric. Following prior work (Guo et al., 2017), predictions are partitioned into $M$ equally spaced confidence intervals $(B_1, B_2, \cdots, B_M)$, where $B_m = \left\{i \in \mathcal{V} \mid \frac{m-1}{M} < \tilde{p}_i \leq \frac{m}{M}\right\}$ and $\tilde{p}_i$ denotes the predictive confidence of node $i$. Following prior work, ECE is evaluated on the ID test nodes only. Let $\mathcal{V}_{ID}^{test}$ test denote the set of ID test nodes. The expected calibration error (ECE) is then defined as $\text{ECE} = \sum_{m=1}^{M} \frac{|B_m|}{|\mathcal{V}_{ID}^{test}|} |\text{acc}(B_m) - \text{conf}(B_m)|$, where $\text{acc}(B_m) = \frac{1}{|B_m|} \sum_{i \in B_m} \mathbb{1}(\tilde{y}_i = y_i)$, $\text{conf}(B_m) = \frac{1}{|B_m|} \sum_{i \in B_m} \tilde{p}_i$.

### 3.2 Deep Reinforcement Learning

Reinforcement learning plays an important role in decision-making processes, and one of its representative frameworks is the Markov Decision Process. A typical MDP can be formulated as $\mathcal{M} = (\mathcal{S}, \mathcal{A}, P, r, \gamma, \rho_0)$, where $\mathcal{S}$ denotes the state space, $\mathcal{A}$ denotes the action space, $P(s' \mid s, a)$ is the state transition probability, $r$ is the reward function, $\gamma \in (0, 1)$ is the discount factor, and $\rho_0$ is the initial state distribution over $\mathcal{S}$. The goal of off-policy reinforcement learning is to learn a policy $\pi(a \mid s)$ that maximizes the expected discounted cumulative reward, $J_\pi = \mathbb{E}\left[\sum_{t=0}^{\infty} \gamma^t r(s_t, a_t)\right]$, by training on data generated from a different behavior policy rather than the target policy. One of the most well-known off-policy methods in deep reinforcement learning is deep Q-learning (Mnih et al., 2013; Van Hasselt et al., 2016). The basic idea of deep Q-learning is to approximate the action-value function $Q(s, a)$ using deep neural networks. The policy is then derived by selecting the action that maximizes the estimated Q-value $a = \arg\max_a Q(s, a) = \arg\max_a \mathbb{E}_{s' \sim P(\cdot \mid s, a)}\left[r + \gamma \max_{a'} Q(s', a')\right]$.

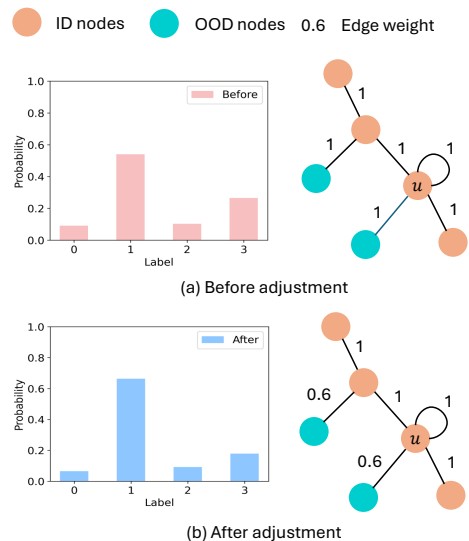

Figure 3: The change of logit distribution before and after adjustment of edge weight. The result is yielded by GCN on Cora.

Apart from Q-value–based methods, which obtain actions implicitly from the Q-function, policy gradient methods (Haarnoja et al., 2018; Wang et al., 2017; Cobbe et al., 2021; Barth-Maron et al., 2018; Tkachenko, 2015; Silver et al., 2014b; Mnih et al., 2016) instead aim to learn the policy directly via a parameterized function $\pi_\theta(a \mid s)$. Similar to deep Q-learning (Mnih et al., 2013;

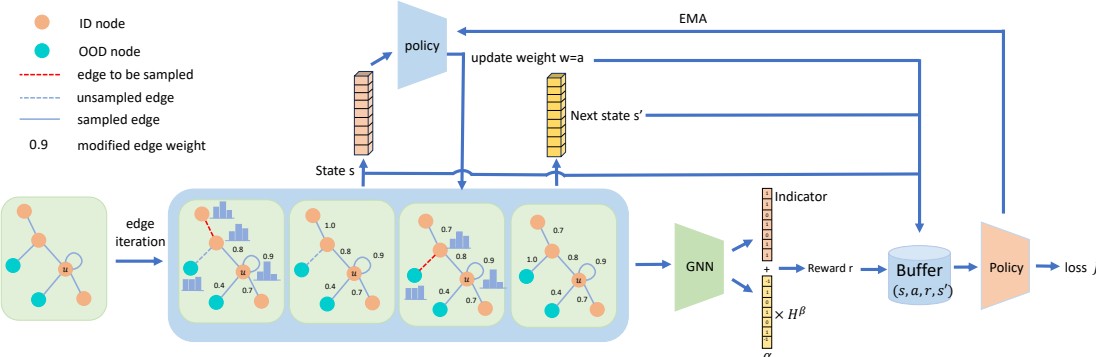

Figure 4: An illustration of our proposed GCSO framework. The method consists of four steps. First, we iteratively traverse adjacent edges from a candidate edge set. Initially, only self-loop edges are considered. At each step, a new edge is sampled from the subgraph without replacement according to a discrepancy score, and a corresponding state is formed. Second, the adjusted edge weight is computed based on the current state and assigned to the newly sampled edge. Next, a reward $r$ is obtained from the GNN with the updated edge weight, and a new state is formed accordingly. The transition tuple is then stored in the replay buffer. Finally, we adopt the DDPG method to train the policy function.

Table 1: Comparison between GCN with original and modified edge weights in terms of node classification accuracy (Acc%)↑ and expected calibration error (ECE%)↓. The experiments are repeated 10 times and the average results are reported. The bold represents the best results.

| Edge weight | Cora | | Citeseer | | PubMed | | CS | | Computers | | Arxiv | |
|---|---|---|---|---|---|---|---|---|---|---|---|---|
| | Acc | ECE | Acc | ECE | Acc | ECE | Acc | ECE | Acc | ECE | Acc | ECE |
| Original | 84.18 | 9.90 | 71.57 | 5.41 | 92.11 | 1.89 | 91.96 | 2.97 | 90.81 | 3.42 | 42.47 | 6.14 |
| Modified | **84.50** | **9.14** | **71.75** | **4.98** | **92.24** | **1.27** | **92.68** | **2.73** | **91.07** | **3.02** | **42.93** | **5.35** |

Van Hasselt et al., 2016), these methods update the parameter $\theta$ of the policy function to maximize the expected discounted cumulative reward. Moreover, modern off-policy policy gradient methods (Haarnoja et al., 2018; Wang et al., 2017; Cobbe et al., 2021; Barth-Maron et al., 2018; Tkachenko, 2015) adopt the actor–critic framework, which models both the policy and the Q-function to achieve improved learning efficiency and convergence. The parameter $\theta$ of the policy function can be updated according to the Policy Gradient Theorem (Sutton et al., 1999):

$$\nabla_\theta J(\theta) = \mathbb{E}\pi[\nabla\theta \log \pi(a \mid s, \theta), Q_\pi(s, a)]. \tag{1}$$

## 4 Empirical Study

In this section, we investigate whether the calibration error of GNNs can be reduced by adjusting edge weights in graphs that include out-of-distribution nodes. Following previous works (Zhao et al., 2020; Stadler et al., 2021), we partition the nodes into ID and OOD categories and adopt GCN (Kipf & Welling, 2016) as the target model. Assuming that the labels and distributions of all nodes are known, we manually modify the weights of edges connected to OOD nodes (e.g., reducing them from 1.0 to 0.6). Experiments are conducted on six benchmark datasets, with details provided in Table 5. The results in Table 1 show that reducing the weights of edges linked to OOD nodes effectively decreases the calibration error while maintaining comparable node classification accuracy relative to using the original edge weights. Our intuition is that adjusting edge weights regularizes the entropy of node predictions, thereby altering the model's confidence without sacrificing accuracy. To validate this hypothesis, we examine the output distributions of GCN (Kipf & Welling, 2016) on the Cora dataset (Yang et al., 2016) before and after modifying the edge weights. As shown in Fig. 3, the results indicate a slight shift in predictive confidence, while the predicted labels remain unchanged. These findings motivate us to design new methods for learning edge weights that improve the calibration of GNNs in the presence of OOD nodes.

## 5 Methodology

In this section, we provide an overview of our framework. We first introduce the formulation of the edge iteration process and define the key components of our method, including the state, action, and reward. We then present the details of the training pipeline. Finally, we discuss our method and provide an analysis of its time complexity.

### 5.1 Iterative Edge Sampling and Re-weighting

We first construct a candidate node set $\mathcal{I}$ by sampling from the node set $\mathcal{V}$. For each target node $u \in \mathcal{I}$, we consider its adjacent edges $\mathcal{E}^u = \{e_0^u, e_1^u, \cdots, e_{k-1}^u\}$, where $e_0^u$ denotes the self-loop edge. Our goal is to learn calibration-friendly edge weights for these edges. To this end, we formulate the edge reweighting process as a Markov Decision Process. Starting from the self-loop edge at time step $t = 0$, the agent iteratively samples a new edge from $\mathcal{E}^u$, assigns it a weight, and observes the resulting calibration reward. Through this sequential decision-making process, the policy gradually learns how different edges influence the calibration of the target node and generates an optimized graph structure.

For example, for a target node $u$, the trajectory starts from its self-loop edge and sequentially processes sampled adjacent edges. At each step, the policy assigns a weight to the current edge and receives a reward. After all sampled edges have been processed, the collected transitions are used to update the actor and critic networks. The corresponding state, action, and reward are defined as follows.

**State**. The state summarizes the information of the edges that have been selected so far around the target node. The state $s_t \in \mathcal{S}$ at time step $t$ in our framework is defined as:

$$s_t = h(s_{t-1}, f_{e_t}), \tag{2}$$

where $f_e$ denotes the feature of edge $e$, and $h$ is a function that maps the previous state and the new edge feature to the updated state. We define the edge feature as the average of the features of the two nodes connected by edge $e$. At time step $t = 0$, $s_0 = \mathbf{X}_u$.

In our study, we adopt a moving average scheme to construct the state. Specifically, the state at time step $t$ is given by:

$$s_t = \lambda f_{e_t} + (1 - \lambda)s_{t-1}, \tag{3}$$

where $\lambda$ is a hyperparameter that controls the contribution of the new edge feature to the state.

In each iteration, a new edge is selected from the candidate edge set $\mathcal{E}^u$ according to a discrepancy score $\delta$. The discrepancy score $\delta$ measures the correlation between an adjacent edge $e^u$ and the target node $u$. It is defined as:

$$\delta := \frac{1}{2} \left( \mathrm{KL}(z_1 \parallel z_u) + \mathrm{KL}(z_2 \parallel z_u) \right), \tag{4}$$

where $z_1$, $z_2$, and $z_u$ denote the class prediction distributions of the two nodes connected by edge $e^u$ and the target node $u$, respectively, obtained by applying the softmax function to the corresponding GNN output logits. KL denotes the Kullback–Leibler divergence.

Intuitively, the discrepancy score measures how different the prediction distribution of a neighboring node is from that of the target node. During each iteration, discrepancy scores are updated based on the current edge weights, and the next edge is selected from the remaining unsampled edges. This strategy helps the agent identify edges that have the greatest impact on calibration and learn more effective graph structures.

**Action**. The action corresponds to assigning a weight to the newly sampled edge. A larger weight allows the edge to contribute more strongly to message passing, while a smaller weight suppresses its influence. Since the action space is continuous, i.e., $\mathcal{A} \subseteq (0, 1]$, we adopt a policy function to generate the adjusted edge weight based on the state $s_t$. At time step $t$, the edge weight $w_{e_t}$ for edge $e_t$ is given by:

$$w_{e_t} = \pi(s_t \mid \theta^\pi), \tag{5}$$

where $\pi(\cdot \mid \theta^\pi)$ denotes the policy function, which can be implemented as a neural network with a Sigmoid activation function in the final layer to ensure that the output lies within $(0, 1]$.

**Reward**. The reward is designed to adjust the confidence of the target node according to its calibration status. Specifically, if a confidence bin is overconfident (i.e., confidence exceeds accuracy), the reward encourages higher predictive entropy to reduce confidence. Conversely, if a bin is underconfident (i.e., accuracy exceeds confidence), the reward encourages lower entropy to increase confidence. For incorrectly classified nodes, the reward encourages the policy to move away from the current prediction distribution.

Formally, let $\tilde{y}_i$ and $y_i$ denote the predicted and ground-truth labels of node $i$, respectively, and let $H_i$ be the predictive entropy. If the predictive confidence of node $i$ falls into bin $B_m$, the reward is defined as

$$r(s, a) = \mathbb{1}(\tilde{y}_i = y_i) + \alpha H_i^{\beta}, \tag{6}$$

where

$$\begin{cases} \alpha = +1, \beta = 1 & \text{if } \tilde{y}_i = y_i \text{ and } \mathrm{acc}(B_m) < \mathrm{conf}(B_m), \\ \alpha = -1, \beta = 1 & \text{if } \tilde{y}_i = y_i \text{ and } \mathrm{acc}(B_m) > \mathrm{conf}(B_m), \\ \alpha = 0, \beta = 0 & \text{if } \tilde{y}_i = y_i \text{ and } \mathrm{acc}(B_m) = \mathrm{conf}(B_m), \\ \alpha = 1, \beta = 1 & \text{if } \tilde{y}_i \neq y_i. \end{cases} \tag{7}$$

The reward follows three intuitive principles: increase entropy for overconfident predictions, decrease entropy for underconfident predictions, and encourage exploration for incorrect predictions. For overconfident nodes, higher entropy reduces excessive confidence by producing a less concentrated predictive distribution. For underconfident nodes, lower entropy increases confidence by sharpening the predictive distribution. For incorrectly classified nodes, the reward encourages the policy to move away from the current prediction distribution and explore alternative graph structures.

## 5.2 Details of Algorithm

The framework of our proposed method is illustrated in Fig. 4. The overall framework consists of four steps. In the first step, we construct the candidate node set $\mathcal{I}$ from the training and validation nodes. For each candidate node, we iteratively sample adjacent edges and construct the corresponding states, as described in Sec. 5.1. In the second step, the adjusted edge weight is generated by the policy function $\pi_\theta(s)$. To enhance the exploration capability of the policy function in the continuous action space, we reformulate the adjusted edge weight as:

$$w_{e_t}^* = \pi(s_t \mid \theta^\pi) + \epsilon, \tag{8}$$

where $\epsilon$ is noise sampled from a uniform distribution, with an upper bound determined by $\epsilon_0(1 + \frac{t}{T})^{-d}$. Here, $\epsilon_0$ denotes the initial noise level and is empirically set to 0.1, which serves as the maximum value of $\epsilon$, $T$ is the total number of iterations, and $d > 0$ is the decay rate. The decaying schedule gradually reduces exploration noise as training progresses.

In the next step, we obtain the reward $r$ from the GNN backbone according to Eq. 6, and store the transition tuple $(s_t, a_t, r_t, s_{t+1})$ in the replay buffer $\mathcal{B}$. Finally, we adopt the Deep Deterministic Policy Gradient (DDPG) (Lillicrap et al., 2016) method to train the policy function. Similar to deep Q-learning (Mnih et al., 2013), the objective of the critic network $Q(s_t, a_t \mid \theta^Q)$ is to approximate the discounted cumulative reward for a state–action pair by minimizing the loss:

$$L(\theta^Q) = \mathbb{E}_{s_t \sim \mathcal{S}, a_t \sim \mathcal{A}} \left[ \left( Q(s_t, a_t \mid \theta^Q) - y_t \right)^2 \right], \tag{9}$$

where the target value $y_t$ is derived from the Bellman equation (Sutton & Barto, 2018) $y_t = r(s_t, a_t) + \gamma Q(s_{t+1}, \pi(s_{t+1}|\theta^\pi)|\theta^Q)$. Since the policy function deterministically generates continuous edge weights from the state, its parameters can be updated according to the Deterministic Policy Gradient Theorem (Silver et al., 2014a; Lillicrap et al., 2016):

$$\begin{aligned} \nabla_{\theta^\pi} J &= \mathbb{E}_{s_t}[\nabla_a Q(s, a|\theta^Q)_{s=s_t, a=\pi(s_t|\theta^\pi)} \nabla_{\theta^\pi} \pi(s|\theta^\pi)_{s=s_t}] \\ &\approx \frac{1}{N} \sum_i (\nabla_a Q(s, a|\theta^Q)_{s=s_i, a=\pi(s_i|\theta^\pi)} \nabla_{\theta^\pi} \pi(s|\theta^\pi)_{s=s_i}). \end{aligned} \tag{10}$$

The detailed procedures of our proposed method are summarized in Algorithm 1 in the Appendix.

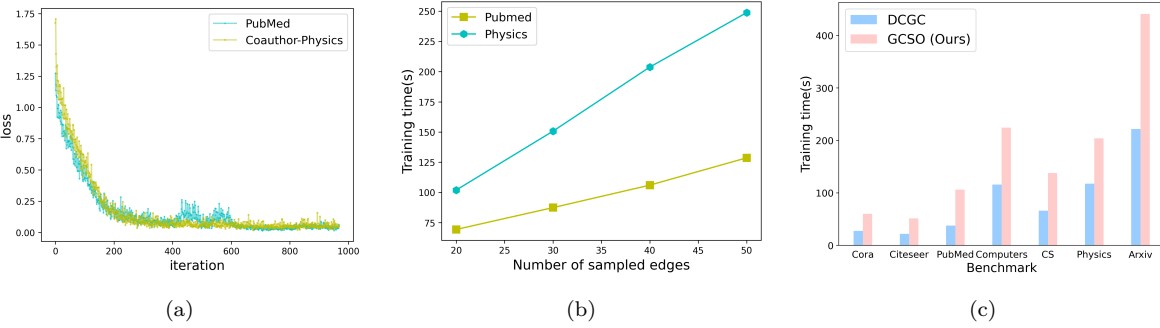

Figure 5: (a) Training curves of our method on PubMed and Coauthor-Physics. (b) Training time of our method under different sampled edge sizes. (c) Comparison of training time between the baseline (DCGC) and our method across different benchmarks. All experiments are conducted on an NVIDIA RTX A5000.

## 5.3 Time Complexity

The main computational cost of our method arises from two components: (1) training the graph neural network at the beginning of each epoch, and (2) training the actor–critic networks at each iteration step. Let $L$ denote the number of layers in both the graph neural network and the actor–critic networks, $|\mathcal{E}|$ the total number of edges, $N$ the number of nodes, $d$ the feature dimension, and $|\mathcal{I}|$ the number of target nodes. The time complexity of the graph neural network is $O(L(|\mathcal{E}|d + Nd^2))$, while the time complexity of the actor–critic networks is $O(L|\mathcal{I}||\mathcal{E}^u|d^2)$. Therefore, the overall time complexity of our framework is $O(L(|\mathcal{E}|d + (N + |\mathcal{I}||\mathcal{E}^u|)d^2))$. According to this analysis, the primary factor influencing the training time is the number of iterations in each trajectory, which is determined by the number of sampled edges. As shown in Fig. 5(b), the training time increases proportionally with the number of sampled edges. Therefore, selecting an appropriate number of sampled edges is crucial for maintaining a reasonable computational cost. In our experiments, we use 10 sampled edges for Cora and Citeseer (Yang et al., 2016), and 40 for the other datasets. Fig. 5(c) presents the actual training time of the data-centric method DCGC (Yang et al., 2024a) and our proposed method across different benchmarks. The results indicate that, although our method requires more training time than the baseline, it maintains a reasonable computational cost even on large graphs. We further evaluate the training dynamics. As shown in Fig. 5(a), our method achieves fast convergence during training. This rapid convergence further reduces the overall computational cost and enhances the scalability of our method to large-sized graphs.

## 5.4 Discussion

First, we discuss the generalization ability of our method. Our policy function is trained on selected edges within a graph. Since there is no distribution shift within a single graph, the learned policy can generalize to other edges in the same graph. However, due to differences in topological structures across graphs, the policy functions for different graphs need to be trained separately. Another advantage of our method is that the optimized graph structure can be readily integrated with existing calibration frameworks (e.g., CaGCN (Wang et al., 2021) and GATS (Hsu et al., 2022)) to achieve improved calibration performance.

Moreover, prior work (Zhang et al., 2020) identifies three desirable properties for calibration methods: **accuracy-preserving**, **data-efficient**, and **expressive**. Our method satisfies these properties. Since the modified graph structure may affect the quality of learned representations and, consequently, downstream task performance (e.g., node classification), we incorporate an indicator function into the reward signal. This component empirically encourages the preservation of classification accuracy on ID nodes during graph structure optimization. In addition, our method is data-efficient. We adopt GCN (Kipf & Welling, 2016) as the GNN backbone and lightweight MLPs for the actor and critic models. Finally, our method is expressive, as it produces fine-grained, adaptive weights for each edge.

Finally, our work is related to a prior study (Shi et al., 2023) that also addresses the calibration of graph neural networks in out-of-distribution scenarios. Although both approaches utilize reinforcement learning, the prior work (Shi et al., 2023) relies on prior knowledge or multiple trials to determine fixed edge weights

Table 2: Comparison between our proposed method and baselines in terms of node classification accuracy (Acc%)↑ and expected calibration error (ECE%)↓ on Cora, Citeseer, PubMed and Chameleon. The experiments are repeated 10 times and the average results and standard deviation are reported. Note that the primary focus of our study is the ECE performance of the methods.

| Methods | Cora | | Citeseer | | PubMed | | Chameleon | |
|---|---|---|---|---|---|---|---|---|
| | Acc | ECE | Acc | ECE | Acc | ECE | Acc | ECE |
| GCN Kipf & Welling (2016) | $84.18 \pm 0.28$ | $9.90 \pm 0.61$ | $71.57 \pm 0.73$ | $5.41 \pm 1.51$ | $92.11 \pm 0.17$ | $1.89 \pm 0.56$ | $47.27 \pm 1.27$ | $14.81 \pm 2.14$ |
| GCL Wang et al. (2022) | $84.19 \pm 0.25$ | $10.05 \pm 0.63$ | $71.91 \pm 0.96$ | $6.07 \pm 2.03$ | $92.14 \pm 0.14$ | $1.76 \pm 0.25$ | $45.81 \pm 0.72$ | $13.98 \pm 1.79$ |
| OODGAT Song & Wang (2022) | $83.17 \pm 1.34$ | $13.96 \pm 3.87$ | $61.95 \pm 0.78$ | $8.52 \pm 2.08$ | $87.44 \pm 0.91$ | $4.64 \pm 1.28$ | $39.74 \pm 4.57$ | $11.06 \pm 5.90$ |
| HyperU-GCN Yang et al. (2022) | $81.88 \pm 1.09$ | $8.40 \pm 7.75$ | $71.27 \pm 1.39$ | $19.69 \pm 13.74$ | $92.35 \pm 0.48$ | $3.24 \pm 0.77$ | $47.36 \pm 2.29$ | $15.22 \pm 9.65$ |
| CaGCN Wang et al. (2021) | $84.14 \pm 0.35$ | $3.85 \pm 1.05$ | $71.57 \pm 0.73$ | $4.47 \pm 0.62$ | $92.11 \pm 0.17$ | $3.09 \pm 0.20$ | $47.27 \pm 1.27$ | $14.15 \pm 1.24$ |
| GATS Hsu et al. (2022) | $83.49 \pm 0.31$ | $2.81 \pm 0.82$ | $72.04 \pm 0.46$ | $5.05 \pm 1.86$ | $92.56 \pm 0.24$ | $2.27 \pm 0.39$ | $46.76 \pm 2.73$ | $11.18 \pm 3.62$ |
| GERDQ Shi et al. (2023) | $83.67 \pm 0.48$ | $9.54 \pm 0.50$ | $69.98 \pm 0.55$ | $4.56 \pm 0.92$ | $92.14 \pm 0.20$ | $1.60 \pm 0.59$ | $46.87 \pm 1.54$ | $14.22 \pm 1.65$ |
| GETS Zhuang et al. (2025) | $84.53 \pm 0.24$ | $3.49 \pm 0.97$ | $\mathbf{72.57} \pm 0.75$ | $5.01 \pm 1.01$ | $92.52 \pm 0.08$ | $1.93 \pm 0.27$ | - | - |
| DCGC Yang et al. (2024a) | $83.91 \pm 0.25$ | $10.44 \pm 0.76$ | $65.02 \pm 0.65$ | $4.62 \pm 1.07$ | $92.26 \pm 0.16$ | $2.43 \pm 0.44$ | $47.60 \pm 1.22$ | $13.28 \pm 1.42$ |
| DCGC+CaGCN Yang et al. (2024a) | $82.99 \pm 0.33$ | $3.04 \pm 0.66$ | $65.00 \pm 0.68$ | $5.13 \pm 0.73$ | $92.66 \pm 0.16$ | $3.01 \pm 0.37$ | $47.47 \pm 0.79$ | $11.95 \pm 1.09$ |
| DCGC+GATS Yang et al. (2024a) | $83.08 \pm 0.31$ | $2.78 \pm 0.33$ | $64.41 \pm 1.17$ | $5.18 \pm 0.50$ | $92.59 \pm 0.20$ | $1.99 \pm 0.21$ | $\mathbf{50.71} \pm 2.43$ | $13.92 \pm 2.58$ |
| GCSO (Ours) | $\mathbf{84.95} \pm 0.18$ | $9.22 \pm 0.26$ | $71.80 \pm 0.70$ | $4.55 \pm 0.66$ | $92.16 \pm 0.16$ | $\mathbf{1.49} \pm 0.20$ | $46.81 \pm 1.03$ | $13.03 \pm 1.59$ |
| GCSO+CaGCN (Ours) | $84.28 \pm 0.27$ | $\mathbf{2.55} \pm 0.45$ | $71.82 \pm 0.68$ | $\mathbf{4.15} \pm 0.48$ | $92.24 \pm 0.26$ | $2.80 \pm 0.19$ | $47.28 \pm 1.21$ | $13.78 \pm 0.82$ |
| GCSO+GATS (Ours) | $84.20 \pm 0.31$ | $2.63 \pm 0.46$ | $72.24 \pm 0.90$ | $4.20 \pm 0.48$ | $\mathbf{92.69} \pm 0.27$ | $2.13 \pm 0.34$ | $46.80 \pm 1.95$ | $\mathbf{10.07} \pm 1.07$ |

for target edges. In contrast, our method learns an adaptive graph structure through reinforcement learning, enabling it to better account for graph topology. Compared to prior methods such as DCGC (Yang et al., 2024a), which do not explicitly consider OOD nodes, our method produces a more distinguishable distribution of edge weights between ID and OOD nodes. As a result, it is more effective in OOD graph settings.

Table 3: Comparison between our proposed method and baselines on CS, Physics, Computers and Arxiv in terms of node classification accuracy (Acc%)↑ and expected calibration error (ECE%)↓. The experiments are repeated 10 times and the average results and standard deviation are reported. Note that the primary focus of our study is the ECE performance of the methods.

| Methods | Coauthor-CS | | Coauthor-Physics | | Amazon-Computers | | OGB-Arxiv | |
|---|---|---|---|---|---|---|---|---|
| | Acc | ECE | Acc | ECE | Acc | ECE | Acc | ECE |
| GCN Kipf & Welling (2016) | $91.96 \pm 0.72$ | $2.97 \pm 0.13$ | $97.02 \pm 0.21$ | $1.67 \pm 0.14$ | $90.81 \pm 0.53$ | $3.42 \pm 0.68$ | $42.47 \pm 0.67$ | $6.14 \pm 0.79$ |
| GCL Wang et al. (2022) | $91.83 \pm 0.41$ | $2.91 \pm 0.29$ | $97.04 \pm 0.21$ | $1.31 \pm 0.12$ | $90.65 \pm 1.04$ | $3.40 \pm 0.59$ | $42.53 \pm 0.63$ | $6.41 \pm 0.79$ |
| OODGAT Song & Wang (2022) | $90.63 \pm 0.35$ | $4.16 \pm 0.60$ | $93.79 \pm 0.52$ | $3.44 \pm 0.38$ | $90.29 \pm 1.02$ | $4.84 \pm 1.01$ | $42.11 \pm 1.19$ | $11.64 \pm 0.74$ |
| HyperU-GCN Yang et al. (2022) | $90.74 \pm 1.03$ | $2.94 \pm 0.84$ | $96.37 \pm 0.57$ | $1.86 \pm 0.70$ | $90.17 \pm 1.37$ | $5.82 \pm 1.10$ | $36.72 \pm 0.65$ | $13.23 \pm 1.58$ |
| CaGCN Wang et al. (2021) | $89.79 \pm 0.40$ | $4.34 \pm 0.40$ | $97.04 \pm 0.20$ | $1.19 \pm 0.13$ | $88.67 \pm 0.38$ | $2.82 \pm 0.17$ | $41.99 \pm 0.75$ | $4.42 \pm 0.35$ |
| GATS Hsu et al. (2022) | $89.28 \pm 0.47$ | $4.14 \pm 0.40$ | $96.80 \pm 0.34$ | $1.21 \pm 0.21$ | $88.07 \pm 0.43$ | $3.59 \pm 0.68$ | $42.07 \pm 0.79$ | $4.84 \pm 0.36$ |
| GERDQ Shi et al. (2023) | $92.36 \pm 0.50$ | $3.12 \pm 0.25$ | $97.05 \pm 0.23$ | $1.38 \pm 0.23$ | $90.52 \pm 0.42$ | $2.62 \pm 0.49$ | $43.58 \pm 0.60$ | $4.70 \pm 0.48$ |
| GETS Zhuang et al. (2025) | $\mathbf{92.91} \pm 0.11$ | $2.95 \pm 0.19$ | $96.23 \pm 0.04$ | $1.15 \pm 0.12$ | $90.47 \pm 0.26$ | $2.63 \pm 0.84$ | $\mathbf{44.16} \pm 0.31$ | $4.67 \pm 0.41$ |
| DCGC Yang et al. (2024a) | $92.03 \pm 0.30$ | $3.36 \pm 0.20$ | $96.97 \pm 0.17$ | $1.52 \pm 0.17$ | $90.84 \pm 0.66$ | $2.69 \pm 0.43$ | $43.62 \pm 0.54$ | $4.85 \pm 0.39$ |
| DCGC+CaGCN Yang et al. (2024a) | $90.11 \pm 0.46$ | $5.52 \pm 0.48$ | $96.81 \pm 0.25$ | $1.15 \pm 0.15$ | $89.17 \pm 0.47$ | $3.15 \pm 0.58$ | $43.08 \pm 0.77$ | $5.68 \pm 0.41$ |
| DCGC+GATS Yang et al. (2024a) | $90.41 \pm 0.59$ | $5.08 \pm 0.56$ | $96.87 \pm 0.39$ | $1.38 \pm 0.23$ | $88.51 \pm 0.72$ | $3.72 \pm 0.64$ | $42.57 \pm 0.72$ | $5.24 \pm 0.45$ |
| GCSO (Ours) | $91.96 \pm 0.25$ | $\mathbf{2.47} \pm 0.13$ | $\mathbf{97.08} \pm 0.21$ | $1.24 \pm 0.12$ | $\mathbf{90.86} \pm 0.45$ | $\mathbf{2.44} \pm 0.28$ | $43.64 \pm 0.93$ | $4.35 \pm 0.49$ |
| GCSO+CaGCN (Ours) | $90.75 \pm 0.45$ | $4.17 \pm 0.16$ | $97.02 \pm 0.21$ | $\mathbf{1.06} \pm 0.06$ | $88.50 \pm 0.30$ | $2.64 \pm 0.29$ | $42.59 \pm 0.51$ | $\mathbf{4.16} \pm 0.39$ |
| GCSO+GATS (Ours) | $90.28 \pm 0.46$ | $3.92 \pm 0.38$ | $96.78 \pm 0.33$ | $1.15 \pm 0.09$ | $88.49 \pm 0.71$ | $3.33 \pm 0.34$ | $42.89 \pm 0.71$ | $4.37 \pm 0.64$ |

# 6 Experiments

In this section, we first introduce the experimental settings. We then present the main results, along with visualizations of reliability diagrams. Sensitivity analyses on key factors, such as the number of sampled edges and hyperparameters (e.g., the discount factor), are provided in Sec. A.4 in the Appendix. Additional results on OOD detection are presented in Sec. A.5 in the Appendix.

## 6.1 Experimental Settings

In the experiments, we perform semi-supervised node classification and compare the performance of our framework with baseline methods on eight benchmark datasets.

**Datasets**. We adopt eight public benchmark datasets, including Cora, Citeseer, PubMed (Yang et al., 2016), Chameleon (Rozemberczki et al., 2021), Coauthor-CS, Coauthor-Physics, Amazon-Computers (Shchur et al.,

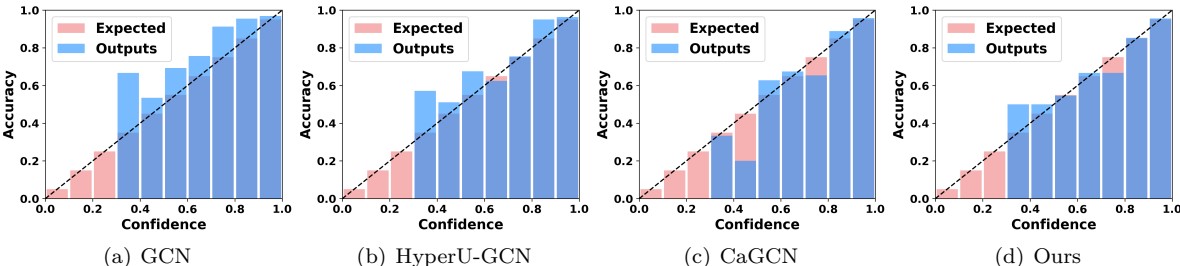

| (a) GCN | (b) HyperU-GCN | (c) CaGCN | (d) Ours |

Figure 6: Reliability diagrams of different methods on Cora with OOD nodes. Well-calibrated results would have closer alignment with the expected results along the diagonal line.

2018), and OGB-Arxiv (Hu et al., 2020). Among these datasets, Chameleon (Rozemberczki et al., 2021) is a heterophilous graph, while the others are homophilous. We follow the train/validation/test splits provided by prior work (Hsu et al., 2022; Yang et al., 2016; Shchur et al., 2018).

To simulate graph learning with OOD nodes, we adopt a label leave-out strategy to partition nodes into ID and OOD sets, following prior work (Gal & Ghahramani, 2016; Song & Wang, 2022; Stadler et al., 2021; Wang et al., 2025). For example, in Cora (Yang et al., 2016), which contains 7 classes, nodes from the last 2 classes are treated as OOD nodes and excluded from the training and validation sets. The remaining nodes are treated as ID nodes. More details about the datasets are provided in Table 5.

**Baselines**. The baseline methods include GCN (Kipf & Welling, 2016), HyperU-GCN (Yang et al., 2022), CaGCN (Wang et al., 2021), GATS (Hsu et al., 2022), GCL (Wang et al., 2022), OODGAT (Song & Wang, 2022), GERDQ (Shi et al., 2023), DCGC (Yang et al., 2024a) and GETS (Zhuang et al., 2025)

**Metrics**. In our experiments, we adopt the expected calibration error (ECE) (Guo et al., 2017) as the primary evaluation metric. A lower ECE indicates better calibration of GNN predictions. In addition, we report node classification accuracy.

**Implementation Details**. We adopt GCN (Kipf & Welling, 2016) as the GNN backbone. The hyperparameters of GCN are set to be consistent with the corresponding baselines. Specifically, the learning rate is $1e-2$, the weight decay is $5e-4$, and the hidden dimension is 64. The actor and critic networks are implemented as three-layer MLPs, with hidden dimensions of 256 and 16, respectively. More implementation details are provided in Sec. A.2 the Appendix.

## 6.2 Main Results

Table 2 and Table 3 present the performance of our proposed method and the baselines on the benchmark datasets. The results show that standard GNN models, such as GCN (Kipf & Welling, 2016), tend to exhibit large calibration errors. Moreover, the results suggest that methods designed for GNN calibration can generally improve calibration performance over GCN (Kipf & Welling, 2016). However, these methods may fail on certain benchmark datasets. For instance, although CaGCN (Wang et al., 2021) achieves low calibration error on Cora (Yang et al., 2016) and Citeseer (Yang et al., 2016), it yields worse calibration performance on PubMed (Yang et al., 2016) and Coauthor-CS (Shchur et al., 2018) compared to GCN (Kipf & Welling, 2016). This phenomenon can be attributed to reduced homophily in the graph, which makes the regularization term in CaGCN (Wang et al., 2021) less effective on these datasets. Additionally, our experimental results show that GERDQ (Shi et al., 2023) and DCGC (Yang et al., 2024a) effectively reduce calibration error across various benchmarks. This further validates that refining graph topology is beneficial for improving calibration, particularly in the presence of OOD nodes.

The results suggest that our proposed GCSO effectively improves the calibration of GNNs and achieves lower ECE than other graph structure optimization methods, such as GERDQ (Shi et al., 2023) and DCGC (Yang et al., 2024a), on most datasets. DCGC (Yang et al., 2024a) can generate an adjusted graph structure to reduce calibration error; however, it does not account for the presence of OOD nodes. GERDQ (Shi et al., 2023) also employs edge reweighting to calibrate GNN predictions, but its adjustments are fixed and not adaptive to the graph topology. In contrast, our approach utilizes a specialized reward signal

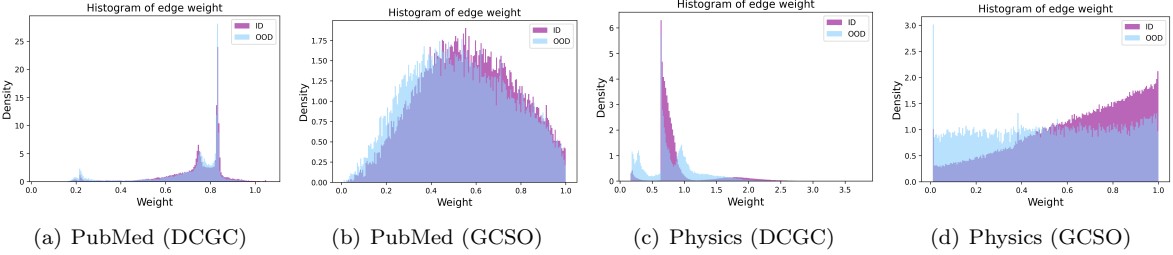

(a) PubMed (DCGC)  (b) PubMed (GCSO)  (c) Physics (DCGC)  (d) Physics (GCSO)

Figure 7: The distribution of edge weights connecting to ID nodes and OOD nodes from DCGC and our proposed method on various datasets. Compared to DCGC, the edge weights generated by our method exhibit a clear distribution shift between ID and OOD nodes.

Table 4: Comparison between our method with complete reward signal, reward signal without entropy and reward signal without indicator function on Cora, PubMed, Coauthor-CS and Amazon-Computers in terms of node classification accuracy (Acc%)↑ and expected calibration error (ECE%)↓.

| Methods | Cora | | PubMed | | Coauthor-CS | | Amazon-Computers | |
|---|---|---|---|---|---|---|---|---|
| | Acc | ECE | Acc | ECE | Acc | ECE | Acc | ECE |
| w/o entropy | $84.88 \pm 0.24$ | $10.01 \pm 0.60$ | $\mathbf{92.16} \pm 0.61$ | $1.75 \pm 0.28$ | $91.80 \pm 0.50$ | $2.92 \pm 0.46$ | $\mathbf{90.91} \pm 0.59$ | $2.80 \pm 0.54$ |
| w/o indicator | $84.05 \pm 0.27$ | $9.58 \pm 0.69$ | $91.62 \pm 0.20$ | $1.57 \pm 0.23$ | $91.22 \pm 0.35$ | $2.51 \pm 0.36$ | $90.08 \pm 0.75$ | $2.50 \pm 0.36$ |
| Complete | $\mathbf{84.95} \pm 0.18$ | $\mathbf{9.22} \pm 0.26$ | $\mathbf{92.16} \pm 0.16$ | $\mathbf{1.49} \pm 0.20$ | $91.96 \pm 0.25$ | $\mathbf{2.47} \pm 0.13$ | $90.86 \pm 0.45$ | $\mathbf{2.44} \pm 0.28$ |

to dynamically assess the impact of adjusted edge weights on target nodes, thereby learning an optimal, adaptive graph structure that regularizes the predictive confidence of GNNs. However,We observe that the calibration gains of GCSO vary across datasets. In particular, the improvements are relatively smaller on Cora and Citeseer, while larger gains are achieved on datasets such as Amazon-Computers and OGB-Arxiv. We believe this behavior is related to the extent to which OOD-node contamination affects message passing and confidence propagation. For smaller citation graphs, the influence of OOD nodes tends to be more localized, and existing calibration methods already achieve strong performance, leaving less room for improvement. In contrast, larger graphs with more complex neighborhood structures are more susceptible to OOD-node interference, allowing graph structure optimization to more effectively reduce harmful interactions and improve calibration. Furthermore, when integrated with existing post-hoc calibration methods, our approach can further enhance calibration performance compared to prior work (e.g., DCGC (Yang et al., 2024a)). To further validate the generalizability and effectiveness of our method in realistic scenarios, we conduct additional experiments under diverse OOD settings on benchmark datasets. The results for both our method and the baselines are presented in Sec. A.3 in the Appendix.

## 6.3 Ablation Study

In our framework, the reward consists of two components: an indicator function and an entropy regularization term. To investigate the contribution of each component, we conduct an ablation study in which only one term is included in the reward. The results are presented in Table 4. The indicator function is designed to encourage the model to maintain comparable accuracy. Without it, our method tends to suffer a drop in accuracy on in-distribution nodes, along with an increase in calibration error. Similarly, removing the entropy regularization term from the reward function reduces the effectiveness of our approach in calibrating graph neural networks. These observations suggest that both components of the reward are indispensable in our framework.

## 6.4 Visualization

To better visualize ECE, the reliability diagrams for our method and the baselines on Cora (Yang et al., 2016) are shown in Fig. 6. Well-calibrated predictions are expected to align closely with the diagonal line. As illustrated in Fig. 6, our method exhibits better alignment with the diagonal compared to the baselines. We further visualize the modified graph structures produced by DCGC (Yang et al., 2024a) and our proposed method on various datasets. More visualizations of reliability diagrams are provided in Sec. A.6 in the

Appendix. The histogram in Fig. 7 shows the distribution of edge weights connecting ID and OOD nodes. Since DCGC does not explicitly account for OOD nodes, the distributions of edge weights for ID and OOD connections exhibit significant overlap. In contrast, our method generates topology-aware edge weights. Fig. 7 demonstrates a clear separation between the edge weight distributions of ID and OOD connections. More illustrations on other datasets are shown in Fig. 13.

## 7 Conclusion

In this paper, we study the calibration of graph neural networks in the presence of OOD nodes, where OOD-node contamination degrades calibration performance. We reveal that OOD nodes can simultaneously induce both overconfidence and underconfidence, a phenomenon that has not been explicitly leveraged by prior graph OOD calibration methods. Motivated by this observation, we propose Graph Calibration via Structure Optimization (GCSO), which learns calibration-friendly graph structures through adaptive edge reweighting. Extensive experiments show that GCSO effectively mitigates the influence of OOD nodes and consistently enhances the performance of existing post-hoc calibration methods. We hope our findings inspire future research on graph calibration under distribution shifts.

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

---

**Algorithm 1** GCSO framework

---

**Require:** Input graph $\mathcal{G} = (\mathcal{V}, \mathcal{E}, X)$, GNN backbone $f$, node labels $Y$, candidate node set $\mathcal{I}$, critic network $Q(s, a \mid \theta^Q)$, actor network $\pi(s \mid \theta^\pi)$, replay buffer $\mathcal{B}$, discount coefficient $\gamma$, hyperparameter $\lambda$, initial noise $\epsilon_0$, total number of episodes $P$, adjacency matrix $\mathbf{A}$
**Ensure:** Refined adjacency matrix $\mathbf{A}'$
1: Initialize the actor network $\pi$, critic network $Q$, and replay buffer $\mathcal{B}$
2: **for** $p = 1$ to $P$ **do**
3:     Train the GNN backbone $f$ with adjacency matrix $\mathbf{A}$ and obtain $\mathrm{acc}(B_m)$ and $\mathrm{conf}(B_m)$ on the validation nodes
4:     Sample a target node $u$ from the candidate node set $\mathcal{I}$
5:     Obtain the edge set $\mathcal{E}^u = \{e_0^u, e_1^u, \ldots, e_{k-1}^u\}$ for node $u$
6:     **for** $t = 1$ to $|\mathcal{E}^u|$ **do**
7:         Calculate the discrepancy scores for unsampled edges according to Eq. 4
8:         Choose the edge $e_t^u$ with the lowest discrepancy score and obtain the state $s_t$ using Eq. 3
9:         Calculate the adjusted edge weight $w_e = a_t$ from state $s_t$ using Eq. 5
10:        Add noise to the adjusted edge weight for exploration via Eq. 8
11:        Assign the adjusted edge weight to the original graph $\mathcal{G}$ to obtain a refined graph structure
12:        Obtain the reward signal $r_t$ from the GNN backbone $f$ via Eq. 6
13:        Store the transition tuple $(s_t, a_t, s_{t+1}, r_t)$ in replay buffer $\mathcal{B}$
14:        Randomly sample data from $\mathcal{B}$ and train the actor network $\pi$ and critic network $Q$ via Eq. 9 and Eq. 10
15:     **end for**
16:     Generate new edge weights and obtain the new adjacency matrix $\mathbf{A}'$ using Eq. 5
17:     Update $\mathbf{A} \leftarrow \mathbf{A}'$
18: **end for**
19: **return** $\mathbf{A}'$

---

Table 5: The statistics of datasets

| Dataset | ID classes | OOD classes | #Nodes | #Edges | #Features |
|---------|-----------|-------------|--------|--------|-----------|
| Cora | [0 - 4] | [5 - 6] | 2,708 | 10,556 | 1,433 |
| Citeseer | [0 - 3] | [4 - 5] | 3,327 | 9,104 | 3,703 |
| PubMed | [0 - 1] | [2] | 19,717 | 88,648 | 500 |
| Chameleon | [0 - 2] | [3 - 4] | 2,277 | 36,101 | 2,325 |
| Coauthor-CS | [0 - 10] | [11 - 14] | 18,333 | 163,788 | 6,805 |
| Coauthor-Physics | [0 - 2] | [3] | 34,493 | 495,924 | 8,415 |
| Amazon-Computers | [0 - 6] | [7 - 9] | 13,752 | 491,722 | 767 |
| OGB-Arxiv | [0 - 29] | [30 - 39] | 169,343 | 1,166,243 | 128 |

# A  Appendix

## A.1  Algorithm

The details of our proposed method are summarized in Algorithm 1.

## A.2  Experimental Setup

**GCN** (Kipf & Welling, 2016): The learning rate is 1e−2 and the weight decay is 5e−4. The model has two layers with a hidden dimension of 64. We use the Adam optimizer (Kingma & Ba, 2014) for training.

**CaGCN** (Wang et al., 2021): CaGCN calibrates GNN confidence via a post-hoc method by modeling different types of uncertainty. In our experiments, we use GCN (Kipf & Welling, 2016) as the base model with a hidden dimension of 64, an initial learning rate of 1e−2, and 8 attention heads.

Table 6: The different OOD configuration of Cora, PubMed and Amazon__Computers.

| Dataset | Config 1 | | Config 2 | | Config 3 | |
|---|---|---|---|---|---|---|
| | ID classes | OOD classes | ID classes | OOD classes | ID classes | OOD classes |
| Cora | [0,3-6] | [1,2] | [0-2,5,6] | [3,4] | [0-4] | [5,6] |
| PubMed | [1,2] | [0] | [0,2] | [1] | [0,1] | [2] |
| Computers | [0,4-9] | [1,2,3] | [0-3,7-9] | [4,5,6] | [0-6] | [7,8,9] |

Table 7: Comparison between our proposed method and baselines in terms of node classification accuracy(Acc%)↑ and expected calibration error(ECE%)↓ on Cora with different OOD configurations.The experiments are repeated 10 times and the average results and standard deviation are reported.

| Methods | Config 1 | | Config 2 | | Config 3 | |
|---|---|---|---|---|---|---|
| | Acc | ECE | Acc | ECE | Acc | ECE |
| GCN (Kipf & Welling, 2016) | $81.91 \pm 0.57$ | $9.88 \pm 0.71$ | $84.85 \pm 0.22$ | $9.42 \pm 0.75$ | $84.18 \pm 0.28$ | $9.90 \pm 0.61$ |
| CaGCN (Wang et al., 2021) | $81.88 \pm 0.50$ | $4.10 \pm 0.44$ | $84.82 \pm 0.26$ | $3.39 \pm 0.57$ | $84.14 \pm 0.35$ | $3.85 \pm 1.05$ |
| GATS (Hsu et al., 2022) | $\textbf{82.61} \pm 0.71$ | $4.11 \pm 0.62$ | $85.42 \pm 0.54$ | $3.03 \pm 1.13$ | $83.49 \pm 0.31$ | $2.81 \pm 0.82$ |
| GERDQ (Shi et al., 2023) | $82.02 \pm 0.59$ | $9.46 \pm 0.60$ | $84.53 \pm 0.46$ | $9.25 \pm 0.35$ | $83.67 \pm 0.48$ | $9.54 \pm 0.50$ |
| DCGC (Yang et al., 2024a) | $80.86 \pm 0.18$ | $8.62 \pm 0.70$ | $83.16 \pm 0.44$ | $8.21 \pm 0.48$ | $83.91 \pm 0.25$ | $10.44 \pm 0.76$ |
| DCGC+CaGCN (Yang et al., 2024a) | $81.11 \pm 0.26$ | $3.03 \pm 0.54$ | $84.19 \pm 0.27$ | $4.62 \pm 0.55$ | $82.99 \pm 0.33$ | $3.04 \pm 0.66$ |
| DCGC+GATS (Yang et al., 2024a) | $80.24 \pm 0.39$ | $2.86 \pm 0.73$ | $83.68 \pm 0.79$ | $4.10 \pm 0.47$ | $83.08 \pm 0.26$ | $2.78 \pm 0.13$ |
| GCSO (Ours) | $81.66 \pm 0.56$ | $9.27 \pm 0.43$ | $84.77 \pm 0.23$ | $9.11 \pm 0.17$ | $\textbf{84.95} \pm 0.18$ | $9.22 \pm 0.26$ |
| GCSO+CaGCN (Ours) | $82.14 \pm 0.42$ | $3.75 \pm 0.38$ | $84.80 \pm 0.28$ | $2.83 \pm 0.38$ | $84.28 \pm 0.27$ | $\textbf{2.55} \pm 0.45$ |
| GCSO+GATS (Ours) | $81.89 \pm 0.54$ | $\textbf{2.49} \pm 0.33$ | $\textbf{85.46} \pm 0.78$ | $2.78 \pm 0.49$ | $84.20 \pm 0.31$ | $2.63 \pm 0.46$ |

**GATS** (Hsu et al., 2022): GATS proposes a temperature scaling technique for calibrating GNNs. We use GCN (Kipf & Welling, 2016) as the base model with a hidden dimension of 64 and a weight decay of 5e−3.

**GCL** (Wang et al., 2022): GCL introduces a loss function for end-to-end calibration of GNNs. The coefficient $\gamma$ is set to 0.020, and the hidden dimension is 64. Other settings follow those of GCN (Kipf & Welling, 2016).

**OODGAT** (Song & Wang, 2022): OODGAT jointly performs node classification and OOD detection. We adopt the same experimental settings as the original work. Note that the ID/OOD split in our experiments differs from that used in OODGAT.

**HyperU-GCN** (Yang et al., 2022): HyperU-GCN focuses on automated graph learning by jointly optimizing model weights and hyperparameters. We follow the original experimental settings.

**GERDQ** (Shi et al., 2023): GERDQ studies GNN calibration in the presence of OOD nodes and mitigates calibration errors by adjusting edge weights via deep Q-learning (Mnih et al., 2013).

**DCGC** (Yang et al., 2024a): DCGC is a data-centric method that improves calibration by assigning larger weights to decisive and homophilic edges.

We carefully select baselines that are most relevant to our problem setting. Specifically, CaGCN (Wang et al., 2021), GATS (Hsu et al., 2022), DCGC (Hsu et al., 2022) and GETS (Zhuang et al., 2025) are representative graph calibration methods widely adopted in the literature. In addition, OODGAT (Song & Wang, 2022) and GERDQ (Shi et al., 2023) are specifically designed to address graph calibration in the presence of OOD nodes, making them particularly relevant baselines for our study. Furthermore, our work is motivated by the observation that OOD nodes introduced through the leave-out setting can simultaneously induce both overconfidence and underconfidence in graph neural networks. To the best of our knowledge, while previous studies have investigated graph calibration and OOD-node settings, there is currently no prior work that explicitly leverages this phenomenon to address graph calibration under OOD-node contamination.

In our method, we adopt GCN (Kipf & Welling, 2016), CaGCN (Wang et al., 2021), and GATS (Hsu et al., 2022) as backbone models. The hyperparameters of GCN follow those used in the corresponding baselines, with a learning rate of 1e−2, weight decay of 5e−4, and hidden dimension of 64.

The actor and critic networks are implemented as three-layer MLPs with hidden dimensions of 256 and 16, respectively. We use the Adam optimizer (Kingma & Ba, 2014) with a learning rate of 1e−3 and weight decay of 1e−2. The parameter $\lambda$ is set to 0.95, and the discount factor $\gamma$ is 0.90. The exploration parameter

Table 8: Comparison between our proposed method and baselines in terms of node classification accuracy(Acc%)↑ and expected calibration error(ECE%)↓ on PubMed with different OOD configurations.The experiments are repeated 10 times and the average results and standard deviation are reported.

| Methods | Config 1 | | Config 2 | | Config 3 | |
|---|---|---|---|---|---|---|
| | Acc | ECE | Acc | ECE | Acc | ECE |
| GCN (Kipf & Welling, 2016) | 83.65 ± 0.27 | 3.95 ± 0.76 | 90.32 ± 0.30 | 1.77 ± 0.74 | 92.11 ± 0.17 | 1.89 ± 0.56 |
| CaGCN (Wang et al., 2021) | 83.67 ± 0.28 | 5.07 ± 0.50 | 90.32 ± 0.31 | 1.64 ± 0.44 | 92.11 ± 0.17 | 3.09 ± 0.20 |
| GATS (Hsu et al., 2022) | 83.47 ± 0.32 | 3.67 ± 0.58 | 89.81 ± 0.30 | 1.47 ± 0.60 | 92.56 ± 0.24 | 2.27 ± 0.39 |
| GERDQ (Shi et al., 2023) | 83.65 ± 0.27 | 3.69 ± 0.57 | 90.32 ± 0.30 | 1.62 ± 0.61 | 92.14 ± 0.20 | 1.60 ± 0.59 |
| DCGC (Yang et al., 2024a) | 83.39 ± 0.41 | 3.68 ± 0.70 | 89.88 ± 0.29 | 2.41 ± 0.33 | 92.26 ± 0.16 | 2.43 ± 0.44 |
| DCGC+CaGCN (Yang et al., 2024a) | 83.26 ± 0.14 | 5.33 ± 0.76 | 90.05 ± 0.31 | 2.67 ± 0.68 | 92.66 ± 0.16 | 3.01 ± 0.37 |
| DCGC+GATS (Yang et al., 2024a) | 82.93 ± 0.62 | 3.59 ± 0.48 | 89.25 ± 0.73 | 2.80 ± 0.68 | 92.59 ± 0.20 | 1.99 ± 0.21 |
| GCSO (Ours) | **83.73** ± 0.28 | 3.68 ± 0.51 | **90.39** ± 0.32 | 1.36 ± 0.43 | 92.16 ± 0.16 | **1.49** ± 0.20 |
| GCSO+CaGCN (Ours) | 83.70 ± 0.26 | 4.86 ± 0.55 | **90.39** ± 0.30 | 1.60 ± 0.21 | 92.24 ± 0.26 | 2.80 ± 0.19 |
| GCSO+GATS (Ours) | 83.52 ± 0.45 | **3.42** ± 0.39 | 89.69 ± 0.40 | **1.30** ± 0.19 | **92.69** ± 0.27 | 2.13 ± 0.34 |

Table 9: Comparison between our proposed method and baselines in terms of node classification accuracy(Acc%)↑ and expected calibration error(ECE%)↓ on Computers with different OOD configurations.The experiments are repeated 10 times and the average results and standard deviation are reported.

| Methods | Config 1 | | Config 2 | | Config 3 | |
|---|---|---|---|---|---|---|
| | Acc | ECE | Acc | ECE | Acc | ECE |
| GCN (Kipf & Welling, 2016) | 87.24 ± 0.58 | 2.74 ± 0.40 | **93.79** ± 0.39 | 2.61 ± 0.23 | 90.81 ± 0.53 | 3.42 ± 0.68 |
| CaGCN (Wang et al., 2021) | 88.14 ± 0.24 | 4.41 ± 0.54 | 92.25 ± 0.42 | 3.16 ± 0.35 | 88.67 ± 0.38 | 2.82 ± 0.17 |
| GATS (Hsu et al., 2022) | 87.50 ± 0.77 | 3.21 ± 0.53 | 92.25 ± 0.49 | 2.75 ± 0.41 | 88.07 ± 0.43 | 3.59 ± 0.68 |
| GERDQ (Shi et al., 2023) | 87.08 ± 0.54 | 2.80 ± 0.23 | 93.41 ± 0.65 | 2.76 ± 0.34 | 90.52 ± 0.42 | 2.62 ± 0.49 |
| DCGC (Yang et al., 2024a) | 87.90 ± 0.36 | 3.32 ± 0.71 | 93.64 ± 0.55 | 2.66 ± 0.41 | 90.84 ± 0.66 | 2.69 ± 0.43 |
| DCGC+CaGCN (Yang et al., 2024a) | 87.73 ± 0.51 | 4.55 ± 0.62 | 93.51 ± 0.67 | 2.81 ± 0.41 | 89.17 ± 0.47 | 3.15 ± 0.58 |
| DCGC+GATS (Yang et al., 2024a) | 87.50 ± 0.51 | 3.78 ± 0.68 | 93.49 ± 0.62 | 2.68 ± 0.36 | 88.51 ± 0.72 | 3.72 ± 0.64 |
| GCSO (Ours) | 87.17 ± 0.41 | **2.56** ± 0.35 | 93.16 ± 0.52 | 2.58 ± 0.16 | **90.86** ± 0.45 | **2.44** ± 0.28 |
| GCSO+CaGCN (Ours) | **88.30** ± 0.56 | 4.03 ± 0.45 | 92.56 ± 0.48 | 2.84 ± 0.19 | 88.50 ± 0.30 | 2.64 ± 0.29 |
| GCSO+GATS (Ours) | 87.65 ± 0.75 | 3.07 ± 0.30 | 92.17 ± 0.35 | **2.40** ± 0.21 | 88.49 ± 0.71 | 3.33 ± 0.34 |

$\epsilon_0$ is set to 0.2. The replay buffer size is 1e4, and the total number of episodes $P$ is 30. We sample 20 target nodes during training and select 10 edges for Cora and Citeseer (Yang et al., 2016), and 40 edges for the remaining datasets. We use 10 bins to evaluate the expected calibration error. All experiments are conducted on an NVIDIA A5000 GPU. We run each experiment 10 times with different random seeds and report the average results.

## A.3 Results under Different OOD Settings

To further assess the generalizability and robustness of our proposed method, we extend our experiments by evaluating it on datasets with different OOD node distributions. Specifically, we conduct evaluations on Cora, PubMed, and Amazon-Computers, each configured with distinct OOD settings. The OOD configurations are detailed in Table 6, and the corresponding results are reported in Tables 7, 8, and 9. The results suggest that conventional methods, such as CaGCN (Wang et al., 2021), struggle to calibrate GNNs under varying OOD configurations. This limitation may stem from the method's lack of consideration of graph topology, particularly the distribution of OOD nodes, when adjusting output logits. In contrast, our proposed method consistently improves calibration across different OOD settings. These improvements are more pronounced on larger graphs (e.g., PubMed and Amazon-Computers) than on smaller graphs (e.g., Cora). Overall, the results across the three datasets demonstrate the strong generalizability and robustness of our approach under diverse OOD distributions.

## A.4 Sensitivity Analysis

We evaluate the impact of the number of sampled target nodes and edges on the performance of our method by conducting additional experiments on the PubMed and Coauthor-CS datasets. In the first experiment, we fix the number of sampled edges to 40 for PubMed and Coauthor-CS, and evaluate the performance using 10, 20,

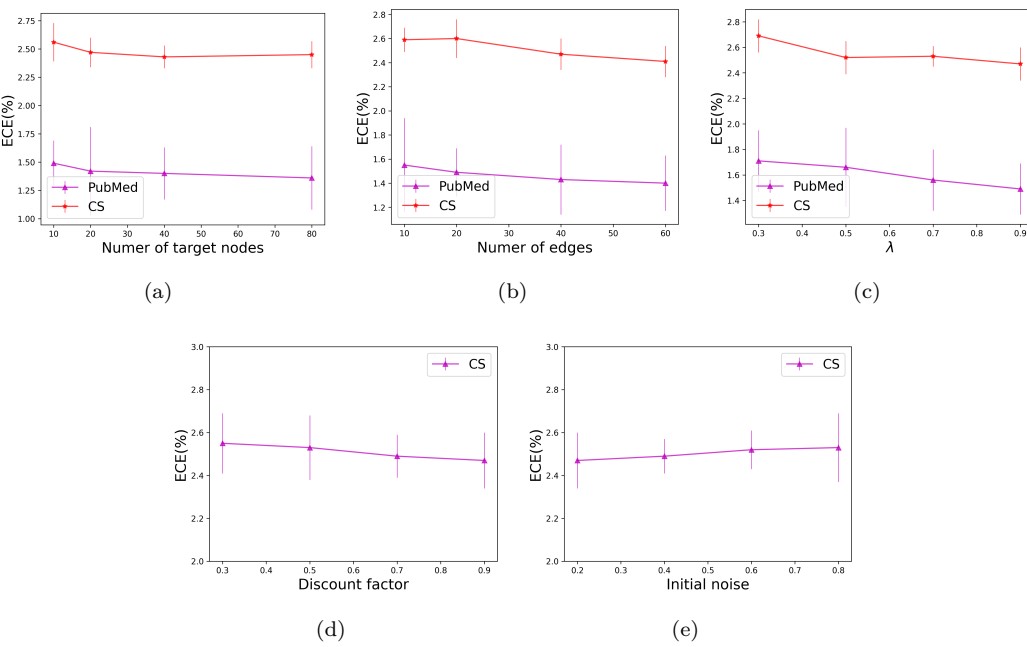

Figure 8: Performance of our method on node classification and calibration on PubMed and Coauthor-CS (a) with varying numbers of labelled nodes, (b) sampled edges and (c) value of $\lambda$. Performance of our method on node classification and calibration on Coauthor-CS with varying value of (d) discounter factor $\gamma$ and (e) initial noise $\epsilon_0$.

Table 10: The performance of GKDE-GCN on node classification and OOD node detection with old and new edge weights. The bold represents the best results.

| Dataset | Edge weight | Acc(%) | ECE(%) | OOD AUROC(%) | OOD AUPR(%) |
|---|---|---|---|---|---|
| PubMed | original | **85.61** | 10.73 | 85.21 | **73.58** |
| | Modified | 85.28 | **9.70** | **85.39** | 72.46 |
| Citeseer | original | 65.43 | 4.56 | 80.75 | 81.72 |
| | Modified | **67.93** | **3.56** | **83.41** | **83.57** |
| Amazon Photo | original | 89.83 | 3.05 | 69.05 | 61.35 |
| | Modified | **91.42** | **1.80** | **69.90** | **62.16** |

40, and 80 sampled nodes. In the second experiment, we fix the number of sampled nodes to 20 for PubMed and Coauthor-CS, and vary the number of sampled edges among 10, 20, 40, and 60. The corresponding results are presented in Fig. 8(a) and Fig. 8(b), respectively. These results indicate that increasing the number of sampled nodes and edges generally improves calibration performance. However, beyond a certain point, the improvements become marginal while the computational cost continues to increase.

We also conduct an evaluation to investigate the influence of the hyperparameter $\lambda$ on the performance of our proposed method. We vary $\lambda$ among 0.3, 0.5, 0.7, and 0.9, and the corresponding results are presented in Fig. 8(c). The results indicate that larger values of $\lambda$ lead to more effective calibration of GNNs. In our method, $\lambda$ controls the update of the state, which is composed of edge features. A smaller $\lambda$ results in less expressive representations, thereby limiting the ability to accurately assess the influence of edges on target in-distribution (ID) nodes.

Finally, we evaluate the sensitivity of our method to reinforcement learning–related hyperparameters, specifically the discount factor $\gamma$ and the initial noise $\epsilon_0$. We vary the discount factor among 0.3, 0.5, 0.7, and 0.9, and the initial noise among 0.2, 0.4, 0.6, and 0.8. The corresponding results are presented in Fig. 8(d) and Fig. 8(e), respectively. The results show that calibration performance tends to degrade when using a smaller discount factor or a larger initial noise value.

## A.5 OOD detection

We conduct a case study to investigate whether the adjusted edge weights can improve the performance of other graph learning methods. GKDE-GCN (Zhao et al., 2020) is a representative method for detecting out-of-distribution (OOD) nodes via uncertainty estimation. We evaluate the performance of GKDE-GCN on both node classification and OOD node detection using the adjusted edge weights learned by our framework. For OOD detection, we report AUROC and AUPR as evaluation metrics. We run the experiments 10 times on Cora, Citeseer, and PubMed (Yang et al., 2016), and report the average results in Table 10. The results show that the adjusted edge weights learned by our method improve both node classification and OOD detection performance of the base model GKDE-GCN.

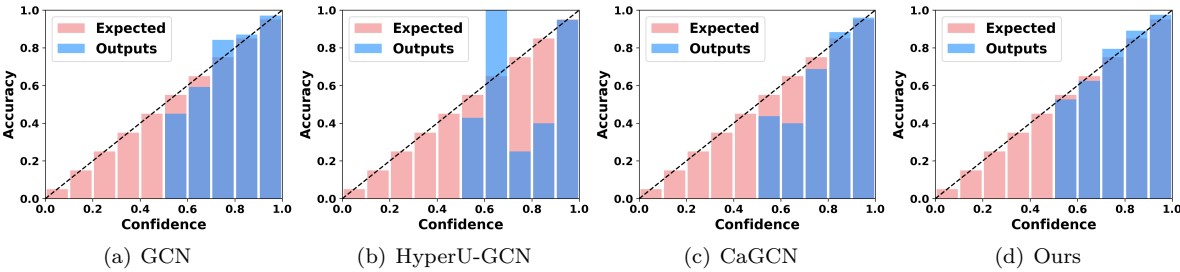

(a) GCN     (b) HyperU-GCN     (c) CaGCN     (d) Ours

Figure 9: Reliability diagrams of different methods on PubMed with OOD nodes. Well-calibrated results would have closer alignment with the expected results along the diagonal line. The results suggest that the calibration issue is different and complicated on different datasets.

## A.6 Visualization

To better visualize the ECE, the reliability diagrams of different methods on different datasets are illustrated from Fig. 9 to Fig. 12.

## A.7 Limitations

Although our proposed method effectively calibrates graph neural networks with out-of-distribution (OOD) nodes through an optimized graph structure, it incurs a higher training time compared to baseline methods due to the inherent complexity of reinforcement learning. Nevertheless, by incorporating strategies such as batch processing and sampling of target nodes and edges, the computational cost remains manageable and does not scale excessively, even on large-scale graphs. Additionally, our current work focuses primarily on calibration for node classification. Extending the approach to other tasks, such as link prediction and graph classification, is an interesting direction for future research.

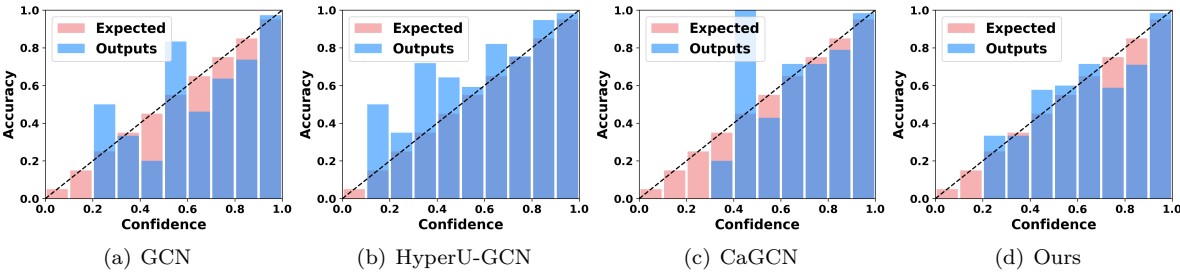

(a) GCN      (b) HyperU-GCN      (c) CaGCN      (d) Ours

Figure 10: Reliability diagrams of different methods on Coauthor-CS with OOD nodes. Well-calibrated results would have closer alignment with the expected results along the diagonal line. The results suggest that the calibration issue is different and complicated on different datasets.

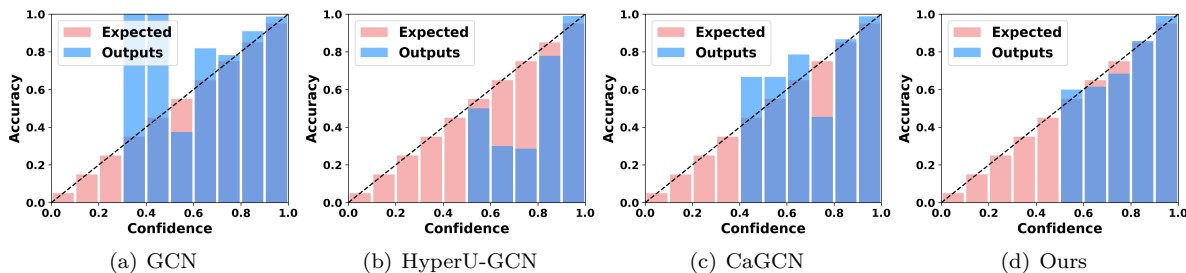

(a) GCN      (b) HyperU-GCN      (c) CaGCN      (d) Ours

Figure 11: Reliability diagrams of different methods on Coauthor-Physics with OOD nodes. Well-calibrated results would have closer alignment with the expected results along the diagonal line. The results suggest that the calibration issue is different and complicated on different datasets.

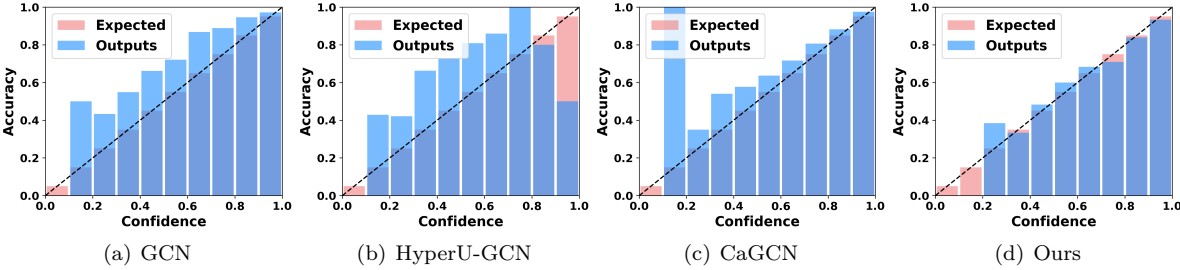

(a) GCN      (b) HyperU-GCN      (c) CaGCN      (d) Ours

Figure 12: Reliability diagrams of different methods on Arxiv with OOD nodes. Well-calibrated results would have closer alignment with the expected results along the diagonal line. The results suggest that the calibration issue is different and complicated on different datasets.

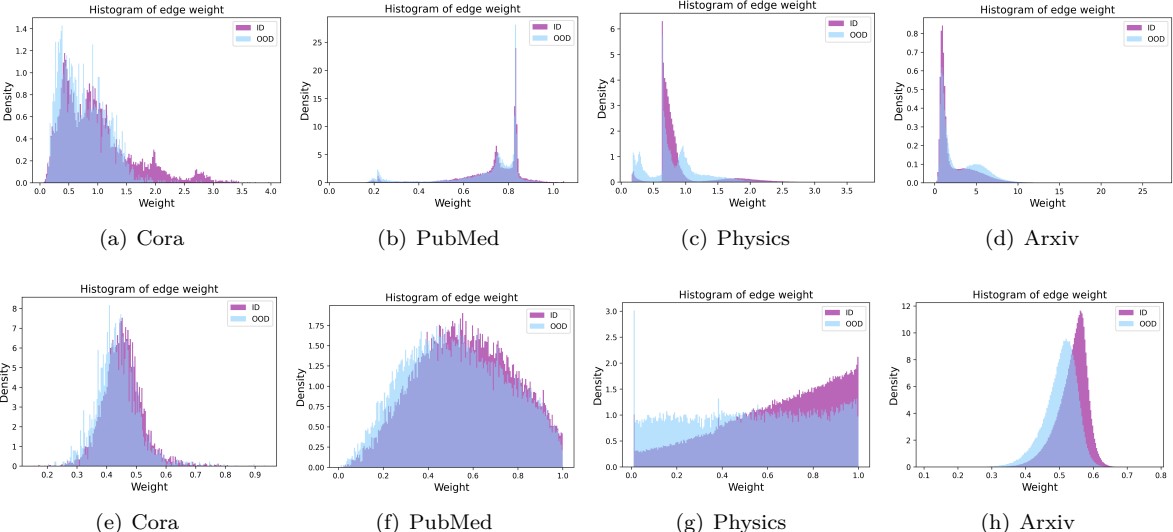

(a) Cora   (b) PubMed   (c) Physics   (d) Arxiv

(e) Cora   (f) PubMed   (g) Physics   (h) Arxiv

Figure 13: The distribution of edge weights connecting to ID nodes and OOD nodes from DCGC ((a)-(d)) and our proposed method ((e)-(h)) on various datasets. Compared to DCGC, the edge weights generated by our method exhibit a clear distribution shift between ID and OOD nodes.

