# OpenReview forum: "Optimized Graph Structures for Calibrating Graph Neural Networks with Out-of-Distribution Nodes"
_TMLR — Decision pending for TMLR_

### Review · Reviewer_jtTZ · 2026-05-12

**Summary Of Contributions:**

This paper studies GNN calibration under OOD-node contamination and proposes GCSO, a DDPG-based edge reweighting method that can also be combined with post-hoc calibration methods. Overall, this paper is strong in the field and has clear evidence to support its claims. Only small issues should be addressed, which could be done in a short period.

Strengths:

1. The problem is important and well motivated.

2. The core idea is reasonable: learning adaptive edge weights is better motivated than using fixed weights, especially when ID-OOD connections vary across graphs.

3. The experiments cover eight datasets and compare against several graph calibration, OOD graph learning, and structure-learning baselines.

Weakness:

1. The calibration target is unclear. OOD nodes are defined as outside the ID label space with unknown labels, but ECE is written over V and requires labels. The paper must clarify whether ECE is computed only on ID test nodes or also on OOD nodes.

2. The results do not fully support the claim that GCSO alone is a strong calibration method. On Cora, GCSO has ECE 9.22, much worse than CaGCN 3.85 and GATS 2.81. On Chameleon, GCSO is also worse than OODGAT and GATS. The strongest results come from GCSO+CaGCN or GCSO+GATS, so the paper should frame GCSO mainly as a structure optimization module that helps post-hoc calibration.

**Audience:**

Yes

**Audience Explanation:**

I believe the paper will likely be of interest to the community, as the OOD and ID problem is important in the graph learning field.

**Claims And Evidence:**

Yes

**Claims Explanation:**

The experimental setup and results are clear, which can support the claim with sufficient evidence.

**Requested Changes:**

1. The paper must clearly define the evaluation set for ECE. Since OOD nodes are outside the ID label space, the authors should state whether ECE is computed on ID test nodes only or on all nodes. If it is ID-only, the task should be described as calibrating ID predictions under OOD-node contamination.

2. The authors should weaken the claim that GCSO alone outperforms existing calibration methods.

3. The authors should include missing relevant calibration baselines, especially methods designed to handle both overconfidence and underconfidence. If a method cannot be applied, the paper should explain why.

---

> ### Author Response · Authors · 2026-05-31
> **Response**
>
> **The calibration target is unclear. The paper must clearly define the evaluation set for ECE.**
>
> Thank you for your valuable suggestion. We apologize for the lack of clarity regarding the evaluation protocol. In our study, the Expected Calibration Error (ECE) is computed **only on the in-distribution (ID) test nodes**, whose ground-truth labels are available. The out-of-distribution (OOD) nodes are treated as unlabeled nodes and are not included in the ECE calculation, since ECE requires access to the true labels to measure the alignment between confidence and accuracy.
>
> Therefore, the goal of our framework is to calibrate the predictions on ID nodes in the presence of OOD-node contamination in the graph structure. The OOD nodes influence message passing and representation learning, which in turn affect the confidence estimates of ID nodes, but they are excluded from the calibration evaluation itself.
>
> To avoid ambiguity, we have revised the manuscript to explicitly state that ECE is evaluated on the ID test nodes only and to clarify the role of OOD nodes in our experimental setting.
>
> **The results do not fully support the claim that GCSO alone is a strong calibration.**
>
> Thank you for your valuable comment. We agree that GCSO alone does not consistently outperform all existing calibration methods across every dataset. In particular, post-hoc methods such as CaGCN and GATS achieve lower ECE than standalone GCSO on some datasets. We apologize for not making this distinction sufficiently clear in the manuscript.
>
> Our primary contribution is not to claim that GCSO is universally superior as a standalone calibration method, but rather to demonstrate that graph structure optimization is an effective way to improve calibration under OOD-node settings. GCSO directly addresses the influence of OOD nodes by learning an adaptive graph structure, which can both improve calibration on its own and serve as a strong foundation for existing post-hoc calibration methods.
>
> Importantly, the experimental results show that combining GCSO with post-hoc calibrators consistently yields stronger performance than applying the same calibrators on alternative graph structures. For example, GCSO+CaGCN and GCSO+GATS achieve lower ECE than their counterparts based on DCGC and other graph-structure optimization approaches. This suggests that the graph structures learned by GCSO are more suitable for calibration under OOD-node contamination.
>
> To better reflect this contribution, we have revised the manuscript to clarify that GCSO should be viewed primarily as a graph structure optimization framework for calibration, whose main strength lies in mitigating the influence of OOD nodes and enhancing the effectiveness of downstream post-hoc calibration methods, rather than as a replacement for existing calibrators.
>
> **The authors should weaken the claim that GCSO alone outperforms existing calibration methods.**
>
> Thank you for the suggestion. We agree that the current wording may overstate the effectiveness of standalone GCSO. Our intention is not to claim that GCSO consistently outperforms all existing calibration methods across all datasets. Instead, GCSO is designed to mitigate the impact of OOD nodes through graph structure optimization, and its strongest results are achieved when combined with post-hoc calibration methods such as CaGCN and GATS.
>
> Accordingly, we have revised the manuscript to moderate the corresponding claims and clarify that the primary contribution of GCSO lies in learning calibration-friendly graph structures under OOD-node settings. We also emphasize that GCSO is complementary to existing post-hoc calibration methods and can further enhance their performance.
>
> **The authors should include missing relevant calibration baselines.**
>
> Thank you for the valuable suggestion. We carefully selected baselines that are most relevant to our problem setting. Specifically, CaGCN, GATS, and DCGC are representative graph calibration methods widely adopted in the literature. In addition, OODGAT and GERDQ are specifically designed to address graph calibration in the presence of OOD nodes, making them particularly relevant baselines for our study.
>
> Furthermore, our work is motivated by the observation that OOD nodes introduced through the leave-out setting can simultaneously induce both overconfidence and underconfidence in graph neural networks. To the best of our knowledge, while previous studies have investigated graph calibration and OOD-node settings, there is currently no prior work that explicitly leverages this phenomenon to address graph calibration under OOD-node contamination.
>
> Therefore, we believe that the selected baselines provide a comprehensive and fair comparison for the problem studied in this paper. To improve clarity, we will revise the manuscript to better justify our baseline selection and further discuss the relationship between our method and existing graph calibration approaches.

---

> > ### Comment · Reviewer_jtTZ · 2026-06-02
> >
> > I appreciate the explanation and revision from the authors. Please integrate them into the final version. I don't have any concerns at this moment.

---

> > > ### Author Response · Authors · 2026-06-02
> > > **Response**
> > >
> > > Thank you for your positive feedback. We will incorporate the corresponding clarifications and revisions into the final version.

---

### Review · Reviewer_nbJg · 2026-05-14

**Summary Of Contributions:**

This paper studies how to calibrate graph neural networks (GNNs) when graphs contain out-of-distribution nodes. The authors propose GCSO, a graph calibration framework that learns adaptive edge weights through structure optimization. GCSO adopts an iterative edge-sampling strategy, formulates edge reweighting as an MDP, and applies an actor–critic method to improve calibration while keeping node classification accuracy competitive. Experiments on multiple benchmark datasets show clear improvements in expected calibration error.
Pros.
(1)The paper addresses an important and underexplored problem: GNN calibration with OOD nodes.
(2)The proposed method is well motivated and clearly explained. Learning adaptive, topology-aware edge weights is a reasonable and effective idea.
(3)The experiments are fairly comprehensive. The paper compares with several strong baselines and reports both calibration and accuracy results.
Cons.
(1)The presentation can be improved. Some parts of the MDP formulation and reward design are dense and would benefit from simpler explanations.
(2)The related work could discuss more recent work on graph OOD detection and uncertainty, such as [a-b], and recent papers on calibration under distribution shift.
[a] GETS: Ensemble Temperature Scaling for Calibration in Graph Neural Networks, ICLR 2025
[b] The Final Layer Holds the Key: A Unified and Efficient GNN Calibration Framework, arXiv, 2025.
(3)Some experimental results need more clarification. For example, it would be helpful to explain why the gains are smaller on some datasets and how sensitive the method is to the number of sampled edges and reward hyperparameters.

**Audience:**

Yes

**Audience Explanation:**

Calibrating GNNs is an important topic in machine learning. Many researchers and practitioners in graph machine learning would be interested in this paper.

**Broader Impact Concerns:**

Please refer to the comments above.

**Claims And Evidence:**

Yes

**Claims Explanation:**

This paper includes comprehensive evaluations on benchmark datasets. The results sufficiently support the claims made in the paper.

**Requested Changes:**

Please refer to the comments above. The authors should cite missing related work, clarify some notations, and add more explanations of the experimental results.

---

> ### Author Response · Authors · 2026-05-31
> **Response**
>
> **The presentation can be improved.**
>
> Thank you for the valuable suggestion. We agree that some parts of the MDP formulation and reward design could be presented more clearly. Our goal was to provide a complete mathematical description of the proposed framework; however, this may have made the presentation overly dense in certain sections.
>
> To improve readability, we will revise the manuscript by providing more intuitive explanations for the MDP formulation and reward design to make the method more accessible.
>
> **The related work could discuss more recent work on graph OOD detection and uncertainty.**
>
> Thank you for the valuable suggestion. We agree that these recent works are highly relevant to our study and should be discussed more thoroughly in the related work section.
>
> In the revised manuscript, we will expand the related work section to discuss these works and clarify the differences between their approaches and ours.
>
> **Some experimental results need more clarification. For example, it would be helpful to explain why the gains are smaller on some datasets and how sensitive the method is to the number of sampled edges and reward hyperparameters.**
>
> Thank you for the helpful suggestion. We agree that the dataset-dependent gains deserve more explanation. Based on Tables 2 and 3, the improvements of GCSO are relatively smaller on Cora and Citeseer, while larger gains are observed on other datasets such as Amazon-Computers and OGB-Arxiv.
>
> We believe gains depend on how much OOD-node contamination affects message passing and confidence propagation. As suggested by previous works [1],  calibration is affected by factors including neighborhood similarity, relative confidence level and the diversity of node-wise predictive distributions. These factors determine how prediction confidence propagates through the graph and how strongly unreliable neighbors can affect a target node.
>
> For smaller citation graphs such as Cora and Citeseer, the graph structure is relatively compact, and the influence of OOD nodes may be more localized. In addition, several post-hoc calibration baselines already achieve strong ECE on these datasets, leaving less room for further improvement. In contrast, larger graphs such as Amazon-Computers contain more nodes and richer neighborhood structures. In these graphs, OOD-node contamination can propagate through more complex neighborhoods and affect a larger number of ID nodes. As a result, topology-aware edge reweighting provides more opportunities to reduce harmful ID-OOD interactions and yields larger calibration gains.
>
> Therefore, the smaller gains on Cora and Citeseer do not necessarily indicate a limitation of the method, but rather reflect that calibration improvements depend on the igraph topology, neighborhood similarity, and the extent of OOD influence. We will revise the manuscript to include this discussion and better explain the dataset-dependent behavior of our method.
>
> To evaluate the impact of the number of sampled edges on the performance of our method, we  have already conducted additional experiments on the PubMed and Coauthor-CS datasets and the results are shown in Sec. A.4. In this evaluation, we fixed the number of sampled nodes to 10 for PubMed and 20 for Coauthor-CS, and varied the number of sampled edges among 10, 20, 40, and 60. The corresponding results are presented in Table 1. These results indicate that increasing the number of sampled edges generally improves calibration performance. However, beyond a certain point, the improvements become marginal while the computational cost increases.
>
> As for the reward hyperparameters., the coefficients alpha and beta in Eq.7 are not treated as tunable hyperparameters. Instead, they are determined by the calibration status of each confidence bin (i.e., overconfident or underconfident) and are selected accordingly. Therefore, the reward is adaptive to different calibration scenarios. The sensitivity of other hyperparameters can be found in Sec. A.4. We will include these discussions in the revised manuscript.
>
> Table 1: Performance of our method on node classification and calibration with varying numbers of edges on PubMed and Coauthor-CS.
>
> | Dataset | \#edge (10) |  | \#edge (20) |  | \#edge (40) |  | \#edge (60) |  |
> |----------|----------|----------|----------|----------|----------|----------|----------|----------|
> |  | Acc | ECE | Acc | ECE | Acc | ECE | Acc | ECE |
> | PubMed | 92.23 ± 0.22 | 1.55 ± 0.39 | 92.16 ± 0.16 | 1.49 ± 0.20 | 92.14 ± 0.28 | 1.43 ± 0.29 | 92.39 ± 0.22 | 1.40 ± 0.23 |
> | Coauthor-CS | 91.94 ± 0.31 | 2.59 ± 0.10 | 91.88 ± 0.36 | 2.60 ± 0.16 | 91.96 ± 0.25 | 2.47 ± 0.13 | 91.93 ± 0.29 | 2.41 ± 0.13 |
>
>
> [1] Hsu, Hans Hao-Hsun, et al. "What makes graph neural networks miscalibrated?." Advances in Neural Information Processing Systems 35 (2022): 13775-13786.

---

### Review · Reviewer_a6dj · 2026-05-17

**Summary Of Contributions:**

This paper introduces a novel framework called Graph Calibration via Structure Optimization (GCSO) to calibrate GNNs in the presence of out-of-distribution (OOD) nodes. GCSO leverages an actor-critic method to dynamically adjust edge weights and evaluate their impact on target node predictions. The research topic is a bit incremental, while the innovation is relatively low.

**Additional Comments:**

None.

**Audience:**

Yes

**Audience Explanation:**

A few people who have interests to the calibration of GNNs may have interests to this work.

**Broader Impact Concerns:**

I don't think that there is any ethical problem in this paper.

**Claims And Evidence:**

Yes

**Claims Explanation:**

The authors cite related papers and use experiments to support their claims in this paper. The experimental results are convincible.

**Requested Changes:**

1.In Section 1, past tense is used. I suggest to use present tense here.
2.The abbreviation needs to be give for the first time it is used. After then, it can be used directly, but has no need to give the abbreviation repeatedly.
3.GraphNAS is a neural architecture search method. In this paper, graph may be a topological structure with nodes and edges. Hence, what’s the relationship between it and the GNNs?
4.In Eq. 4, what’s the logit distributions of the two nodes?
5.In Eq. 7, what’s the range of the entropy H_i?
6.How to derive the upper bound of \epsilon?
7.In the experiments, baseline methods published in 2025 and 2026 should be added and compared with.
8.In Section 6.3, Table 4 is not in the Appendix.

---

> ### Author Response · Authors · 2026-05-31
> **Response**
>
> **In Section 1, past tense is used. I suggest to use present tense here.**
>
> Thank you for the suggestion. We agree and have revised Section 1 accordingly, replacing the past tense with the present tense where appropriate.
>
> **The abbreviation needs to be give for the first time it is used.**
>
> Thank you for the suggestion. We have revised the manuscript to ensure that all abbreviations are defined when first introduced and are used consistently thereafter without repeated definitions.
>
> **GraphNAS is a neural architecture search method. Hence, what’s the relationship between it and the GNNs?**
>
>
> Thank you for the question. We agree that the distinction between graph topology and GNN architectures could be made clearer in the manuscript.
>
> GraphNAS is cited in the related work section as a representative neural architecture search method for automatically designing GNN architectures. In contrast, our work focuses on graph structure optimization, where the graph topology (i.e., nodes and edges) is modified to improve calibration. These two directions are complementary: GraphNAS optimizes the GNN architecture, while our method optimizes the graph topology on which the GNN operates. Since the graph structure directly determines the message-passing process, modifying the topology can significantly affect both prediction performance and calibration.
>
> To avoid confusion, we will revise the related work section to more clearly distinguish GNN architecture optimization from graph structure optimization and clarify the relationship between the two.
>
> **In Eq. 4, what’s the logit distributions of the two nodes?**
>
> Thank you for the question. In Eq. 4, the logit distributions refer to the class prediction distributions produced by the GNN for the two nodes. Specifically, given the output logits of a node, we apply a softmax function to obtain a probability distribution over all classes. The two distributions in Eq. 4 therefore correspond to the predicted class distributions of the two nodes connected by the edge under consideration.
>
> We agree that this notation was not sufficiently clear in the manuscript. To improve readability, we will revise Eq. 4 and explicitly define the logit distributions, including how they are obtained from the GNN outputs.
>
> **In Eq. 7, what’s the range of the entropy $H_i$?**
>
> Thank you for the question. In Eq. 7, $H_i$ denotes the entropy of the predictive distribution for node i, computed from the softmax probabilities of the GNN output. Therefore, its range is $0 \le H_i \le \log C$, where $C$ is the number of classes. The lower bound is achieved when the prediction is completely certain, while the upper bound corresponds to a uniform predictive distribution over all classes. We agree that this definition should be stated explicitly and will revise the manuscript to clarify the range and interpretation of $H_i$.
>
> **How to derive the upper bound of $\epsilon$?**
>
> Thank you for the question. The upper bound of $\epsilon$ in Eq. 8 is not derived from a theoretical analysis. Instead, it is designed as a decaying exploration schedule for the policy function during reinforcement learning. Specifically, a larger $\epsilon$ is used in the early training stage to encourage exploration of different edge weights, and the exploration noise gradually decreases as training proceeds to stabilize the learned policy.
>
> In our experiments, the initial exploration scale $\epsilon_0$ is empirically set to 0.1, which serves as the maximum value of $\epsilon$. Therefore, the empirical upper bound of $\epsilon$ is 0.1 at the beginning of training and decreases according to the decay schedule $\epsilon_0(1+t/T)^{-d}$. We will revise the manuscript to clarify this design.

---

> ### Author Response · Authors · 2026-05-31
> **Response 2**
>
> **In the experiments, baseline methods published in 2025 and 2026 should be added and compared with.**
>
> Thank you for the valuable suggestion. Following the reviewer's recommendation, we additionally compare our method with GETS [1], a recent state-of-the-art graph calibration method based on ensemble temperature scaling.
>
> For a fair comparison, we follow the same experimental protocol and evaluation metrics used in the main paper and report results on multiple benchmarks. Part of the results are shown in Table 1. These results demonstrate that our graph structure optimization strategy remains effective compared with recent graph calibration approaches and can achieve superior calibration performance while maintaining strong predictive accuracy. We will add the new results in our revised manuscript.
>
> [1] Zhuang, Dingyi, et al. "Gets: Ensemble temperature scaling for calibration in graph neural networks." International Conference on Learning Representations. Vol. 2025. 2025.
>
> Table 1: Performance of our method and baselines on node classification and calibration on multiple benchmarks.
>
> | Dataset | Cora |  | Citeseer |  | Computers |  |
> |----------|----------|----------|----------|----------|----------|----------|
> |  | Acc | ECE ↓ | Acc | ECE ↓ | Acc | ECE ↓ |
> | CaGCN | 84.14 ± 0.35 | 3.85 ± 1.05 | 71.57 ± 0.73 | 4.47 ± 0.62 | 88.67 ± 0.38 | 2.82 ± 0.17 |
> | GETS | **84.53 ± 0.24** | 3.49 ± 0.97 | **72.57 ± 0.75** | 5.01 ± 1.01 | 90.47 ± 0.26 | 2.63 ± 0.84 |
> | Ours | 84.28 ± 0.27 | **2.55 ± 0.45** | 71.82 ± 0.68 | **4.15 ± 0.48** | **90.86 ± 0.45** | **2.44 ± 0.28** |
>
> **In Section 6.3, Table 4 is not in the Appendix.**
>
> Thank you for pointing this out. We apologize for the incorrect reference. Table 4 is included in the main paper rather than the Appendix. We have corrected the corresponding citation in Section 6.3 to ensure consistency and avoid confusion.

---

### Decision · Action_Editor_UZZF · 2026-06-25

**Recommendation:** Accept with minor revision

**Additional Comments:**

1. The abstract is not concise. I suggest directly introducing the research question targeted in this paper, rather than starting from GNNs.
2. The abstract mentions that "current graph structure learning methods do not explicitly address the calibration challenges posed by out-of-distribution (OOD) nodes," which is not accurate. There are several existing methods studying the calibration problem. In other words, the authors need to clearly demonstrate the drawbacks of existing methods, where the original word "explicitly" is not enough.
3. I do not suggest putting Figure 1 in the introduction section, which needs a figure for the motivation, rather than the empirical results. Intuitively, the performance degrades with OOD nodes. Moreover, if you want to use the empirical results as the motivation, the detailed settings should be provided. The same applies to Figure 2, which should be moved to the experimental section.
4. The caption of Section 4 "Empirical Study" is weird. I believe it plays the role of motivation.
5. Font sizes among figures are not consistent. Some are too tiny.
6. Page 17 should start with the section "Appendix."
7. Typos. "However,We" on Page 12.

**Audience:**

Yes

**Audience Explanation:**

Graph learning is a long-lasting research topic in the machine learning area, which should attract large audience.

**Claims And Evidence:**

Yes

**Claims Explanation:**

The authors point out that current graph structure learning methods do not explicitly address the calibration challenges posed by out-of-distribution (OOD) nodes. To tackle this challenge, the authors propose a novel framework called Graph Calibration via Structure Optimization to calibrate GNNs in the presence of OOD nodes. The empirical results demonstrate that the proposed method delivers better calibration performance than other methods.

Three reviewers recommended the consensus acceptance, and I also hold the same recommendation with additional comments in terms of logic and accurate presentation. Therefore, I recommend "Accept with minor."